

# Optimal control strategies for parameter estimation of quantum systems

**Quentin Ansel⋆, Etienne Dionis and Dominique Sugny†**

Laboratoire Interdisciplinaire Carnot de Bourgogne, CNRS UMR 6303,
Université de Bourgogne, BP 47870, F-21078 Dijon, France

⋆ quentin.ansel@u-bourgogne.fr , † Dominique.Sugny@u-bourgogne.fr

## Abstract

Optimal control theory is an effective tool to improve parameter estimation of quantum systems. Different methods can be employed for the design of the control protocol. They can be based either on Quantum Fischer Information (QFI) maximization or selective control processes. We describe the similarities, differences, and advantages of these two approaches. A detailed comparative study is presented for estimating the parameters of a spin$-\frac{1}{2}$ system coupled to a bosonic bath. We show that the control mechanisms are generally equivalent, except when the decoherence is not negligible or when the experimental setup is not adapted to the QFI. In this latter case, the precision achieved with selective controls can be several orders of magnitude better than that given by the QFI.

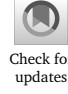

# 1   Introduction

Quantum Metrology using quantum features for parameter estimation has recently attracted increasing attention because it can outperform any classical resource-based measurement scheme [1–8]. Despite impressive precision gains that can be achieved, ultimate performance can only be attained if the various steps of the protocol are optimized [4,9,10]. Standard processes usually consider a free evolution of the system initially prepared in an optimal initial state. However, in many examples, this approach is not sufficient and the system dynamics must be modified by external control to achieve the highest precision for given experimental constraints.

The control design is usually performed by Optimal Control Theory (OCT), which has proven its effectiveness in many quantum applications [6,11–14]. Different solutions have been proposed so far, to define the optimal control problem. They schematically differ in the quantity to be maximized (or minimized) at a fixed final time. Among others, we can mention the maximization of Quantum Fisher Information (QFI) [10,15–30], selective control protocols [31–39] and the fingerprinting method [40–43]. QFI is based on a generalization of the Cramér-Rao bound to quantum systems [9,44,45]. For pure states, the QFI is proportional to the variance of a specific observable, related to the partial derivative of the Hamiltonian with respect to the parameter to estimate. By maximizing this quantity, we ensure that a small perturbation of the parameter induces a significant modification of the system dynamics, and therefore, this allows us to reduce the error made during a measurement. For QFI, the information is local in the parameter space and there is no explicit target quantum state in the definition of the control problem. This is not the case with selective control processes which are non-local by nature. They can be viewed as a simultaneous state-to-state control protocol for different copies of the system characterized by different values of parameters [33,34,36, 46–50]. Selective control has been used extensively in Nuclear Magnetic Resonance [51–55]. In this framework, the goal is to find a control that allows us to reach (possibly as fast as possible) different target states for each copy of the system, the target states being chosen specifically to minimize measurement error. The fingerprinting method is more elaborated and combines ideas coming from QFI and selective protocols [40–43]. There are no specific target states but the goal is to maximize the distance between the time evolution of one or several observables. In this case, the whole dynamic is taken into account, not just the final system configuration [43]. In addition to the maximization of a given figure of merit, other constraints can be included in the analysis of these problems, such as the minimization of control time or energy [56–59].

Different control strategies can be obtained independently with these approaches, e.g., for parameter estimation of spin systems. A question that naturally arises is to know under which conditions these control protocols are equivalent, and more generally the advantages, similarities, and differences between the different techniques. To the best of our knowledge, only the fingerprinting method has been briefly connected to Fisher information in [60,61], but the

relation between QFI and selective protocols remains unexplored. This paper aims at taking a step in this direction. To simplify the analysis, we focus on the link between QFI and selective control protocols for the question of precision measurement. The two techniques may look very different at first glance, but they are deeply related and can be seen as complementary approaches to the same problem. In view of practical applications, we discuss the advantages of the two methods, in terms of precision, ease of implementation, and adaptability with respect to the experimental setup. The analysis is illustrated with a detailed case study, a spin$-\frac{1}{2}$ particle coupled to a bosonic bath at zero temperature. We derive in each case the optimal controls for the estimation of spin frequency, environment damping rate, and scaling factor of the control. For the spin frequency, we recover some of the solutions found independently in the literature [16, 19], but original solutions are also determined in the other cases. To complete the study, we also explore how some of the optimized controls behave in a simulation of an experiment in which the spin frequency is estimated using a maximum likelihood method [62–65].

This article is organized as follows. In Sec. 2 and 3, we introduce several basic notions about QFI and selective control processes, and we present the formal relation between the two approaches. Several state-of-the-art mathematical results are recalled. In Sec. 4, we introduce and discuss the concept of optimal solution in both cases. In Sec. 5, we provide a detailed comparative study with the example of a spin-1/2 particle. We conclude in Sec. 6. Some mathematical proofs are gathered in the appendices.

## 2 Theoretical background: Quantum and classical Fisher informations

In this section, we present the basic ideas underlying the maximization of QFI and the selectivity in a driven quantum system. We also provide several mathematical results regarding the two control problems.

Let $\mathcal{H}$ be a finite-dimensional Hilbert space associated with the system to be measured. The state of the system at time $t$ is a density matrix given by

$$\rho(t) = \sum_{i=0}^{dim\mathcal{H}-1} p_i(t)|\psi_i(t)\rangle\langle\psi_i(t)|, \tag{1}$$

where $|\psi_i(t)\rangle \in \mathcal{H}$ and $p_i$ is the probability to find the system in the state $|\psi_i(t)\rangle$. We assume that the quantities $p_i$ and $|\psi_i\rangle$ depend on a parameter $X \in \mathbb{R}$ to estimate by experimental means. The time evolution of $\rho(t)$ from an initial state $\rho_0$ is described by an evolution (super) operator [66]:

$$\rho_X(t) = \hat{\mathcal{U}}(X, t, u(t))\rho(0), \tag{2}$$

which can give rise to unitary or non-unitary dynamics, and $u(t)$ is a control that allows us to modify the system time evolution. The subscript $X$ in $\rho_X$ is used to explicitly specify that the density matrix depends on the parameter $X$. A key point here is that the initial state is assumed to be independent of the parameter $X$, and the dependence of $\rho_X(t)$ on $X$ is only due to system dynamics.

Before the experiment, we have a prior estimate of the value of the unknown parameter $X$, denoted $X_0$, for which the uncertainty is large. The goal of the experiment is to obtain more precise information about the system, that is, a better estimate of the value of $X$ with less uncertainty. The approach considered in this article is based on the maximum likelihood method [62–65,67]. When applied to a quantum system, the likelihood function gives a notion of distance between $\rho_X$ and the measurement data. More precisely, the larger the likelihood

function, the higher the probability that $\rho_X$ is the state of the system. Therefore, maximizing the likelihood function gives us the most probable value of $X$ which describes the measurement data. The method also makes it possible to determine the uncertainty of the estimated parameter. This uncertainty is related to the distance between two states that can give approximately the same experimental result. Heuristically, it is clear that if $\rho_{X_1}$ and $\rho_{X_2}$ correspond almost to the same quantum state, then it will be very difficult to discriminate between the two values $X_1$ and $X_2$. The goal of the control process is, therefore, to increase the distance between $\rho_{X_1}$ and $\rho_{X_2}$ as much as possible in a given control time. The procedure is schematically illustrated in Fig. 1.

This approach can be qualified as *non-local* since the difference $|X_2 - X_1|$ can be arbitrarily large. Another option is to consider a *local* derivation of the estimation protocol around $X_0$. In this case, a series expansion of the density matrix is performed as follows:

$$\rho_X = \rho^{(0)} + \rho^{(1)}\delta X + \rho^{(2)}\delta X^2 + O(\delta X^3), \tag{3}$$

with $\rho^{(n)} \equiv \dfrac{\partial^n \rho_X}{\partial X^n}\bigg|_{X=X_0}$. This idea allows us to derive a lower bound on the error made on the estimate of $X$, given by the Cramér-Rao bound for quantum systems [68]. For an *unbiased estimator and a single measurement*, the mean square error $\Delta X^2$ is given by:

$$\Delta X^2 \geq \frac{1}{\mathcal{F}}, \tag{4}$$

where $\mathcal{F}$ is the Quantum Fisher Information (QFI) for a density matrix. As can be seen in Eq. (4), the larger the QFI, the lower the uncertainty can be. The maximization of the QFI at a fixed time using an external control is therefore a way to minimize the measurement error [10, 16, 17, 19–21].

We now focus on QFI and we derive several useful results for the computation of optimal controls. The QFI $\mathcal{F}$ for a density matrix can be defined from different approaches [9, 69]. A standard definition is to introduce the symmetric logarithmic derivative operator $L$, given by the relation $\partial_X \rho_X = \frac{1}{2}(L\rho_X + \rho_X L)$, and to express the QFI as [9]

$$\mathcal{F} = \text{Tr}[\rho_X L^2]. \tag{5}$$

An explicit calculation of the operator $L$ enables us to relate $\text{Tr}[\rho_X L^2]$ to the Bures distance between two quantum states [70]. A link with selective control problems can be established from the Bures metric, which is thus used to derive the QFI in this study.

**Definition 1.** *(QFI)* The Quantum Fisher Information for a full rank density matrix is given by [69]:

$$\mathcal{F} = \lim_{\delta X \to 0} \frac{4}{(\delta X)^2} D(\rho_{X_0}, \rho_{X_0+\delta X})^2, \tag{6}$$

where $D(\rho_{X_0}, \rho_{X_0+\delta X})^2$ is the Bures distance, defined as:

$$D(\rho_1, \rho_2)^2 = 2\left(1 - \text{Tr}\left[\sqrt{\sqrt{\rho_1}\rho_2\sqrt{\rho_1}}\right]\right). \tag{7}$$

The Bures distance gives a simple geometric picture of QFI as the distance between the states of two systems with infinitesimal variations of the parameter $X$. Note that this definition is not strictly equivalent to Eq. (5) when the support of the density matrix is modified by a variation of the parameter $X$, which may happen, e.g., when the state is pure. This point has been discussed in [70–73]. In the case of a pure state, we have:

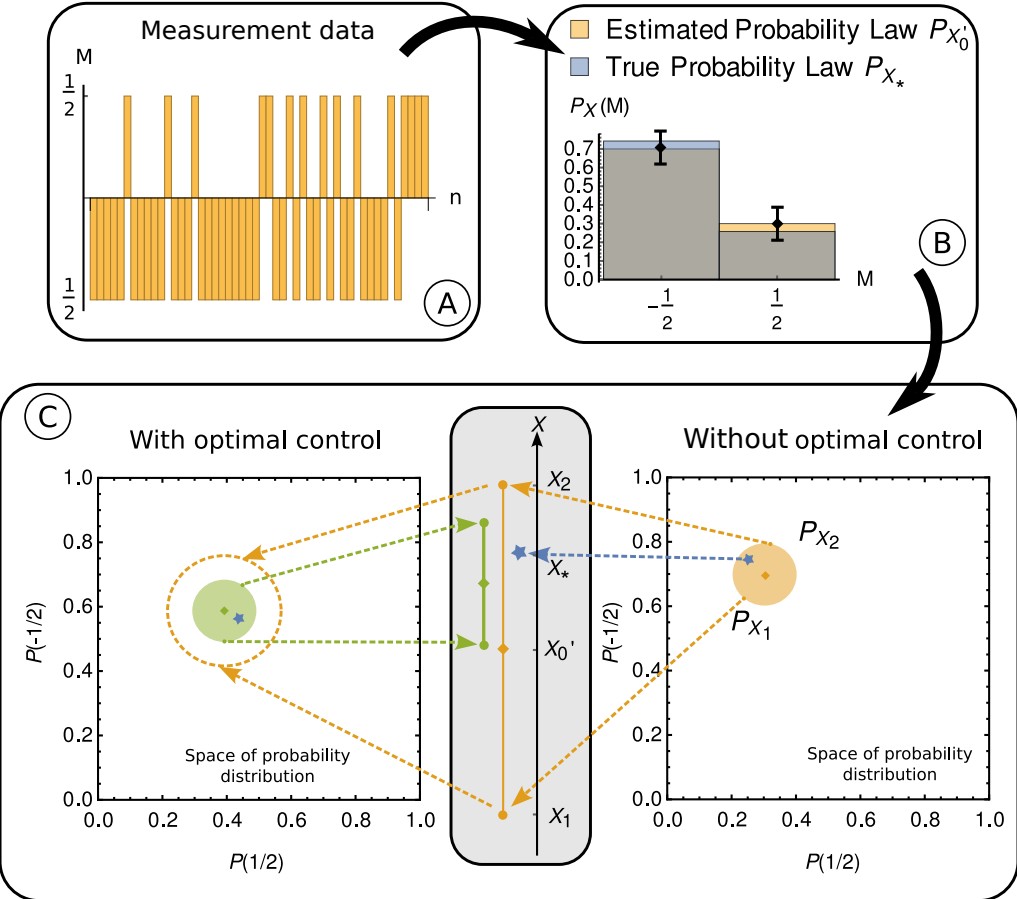

Figure 1: Schematic description of the estimation process of a parameter $X$ using the likelihood method with and without optimal controls. For simplicity of representation, the method is illustrated using only the probability distribution $P$, and not the quantum state $\rho$, but the former is a function of the latter. As shown in panel A, the starting point is a set of independent measurements, obtained after applying a control that prepares the state of the system. From these data, the probability of occurrence for each event can be estimated, as depicted in panel B. The estimated probability is ideally close to "the true" probability law $P_{X_*}$, but there is a necessary small deviation due to the finite number of measurements (the true probability being determined in the limit of an infinite number of measurements). By maximizing the likelihood function and using a bootstrap method [62, 63], we can estimate the parameter $X_0'$ which is the most probable value of $X$ that represents the experimental data (symbolized by colored diamonds). We can also determine the interval of uncertainty, bounded by $X_1$ and $X_2$ (symbolized by colored discs), as shown in panel C. On the right side of panel C, the points representing the different probability laws are obtained for a system driven with a non-optimal control. The role of the optimization process is to design a control protocol that increases the distance between $P_{X_1}$ and $P_{X_2}$, such that for equivalent measurements, with the same uncertainty on the probability distribution, the uncertainty on the parameter $X$ is smaller, as depicted in the left side of panel C.

**Definition 2.** *(QFI of a pure state)* The Quantum Fisher Information for a pure state $\rho(t) = |\psi\rangle\langle\psi|$, with $|\psi\rangle = |\psi^{(0)}\rangle + \delta X|\psi^{(1)}\rangle + O(\delta X^2)$, is given by

$$\mathcal{F} = 4\left(\langle\psi^{(1)}|\psi^{(1)}\rangle - \left|\langle\psi^{(0)}|\psi^{(1)}\rangle\right|^2\right). \tag{8}$$

This corresponds to 4 times the Fubini-Study metric [69], which is equal to the square of the norm of $|\psi_\perp\rangle = |\psi^{(1)}\rangle - \langle\psi^{(0)}|\psi^{(1)}\rangle|\psi^{(0)}\rangle$. This latter vector can be interpreted as the part of $|\psi^{(1)}\rangle$ transverse to $|\psi^{(0)}\rangle$. We stress that the two definitions agree within the appropriate limit as discussed in Appendix A.1.

QFI gives a theoretical minimum limit on measurement precision, but in practice, this limit may not be reached. This is because QFI is related to a specific positive operator-valued measure (POVM) [69], and the experimental setup may not be well suited to this POVM. Additionally, the experiment may not offer an informationally complete POVM [74] which means that the quantum state may not be determined completely by state tomography. As a result, the sensitivity of the measurement could remain low, even if the QFI is maximized. The relevant quantity in the likelihood estimation method is the Classical Fisher Information (CFI) [69], which depends on the POVM of the experimental setup, and not on the *optimal* POVM associated with the QFI. A standard example corresponds to the case where only the state populations in a given basis can be measured (see [75] for an example with Bose-Einstein condensates) whereas QFI may require full tomography of the state, i.e. the measurement of both populations and phases.

**Definition 3.** **(CFI)** Let a POVM, that is a set $\{\Pi_n\}_{n=0,\ldots,d}$, $d \geq dim\mathcal{H}$ of Hermitian and positive semi-definite operators that sum to identity. Let $\pi_n^{(0)} = \text{Tr}[\rho^{(0)}\Pi_n]$ and $\pi_n^{(1)} = \text{Tr}[\rho^{(1)}\Pi_n]$. The Classical Fisher Information, denoted $\mathcal{F}_C$, associated with the quantum state is defined as:

$$\mathcal{F}_C = \sum_{n|\pi_n^{(0)}>0} \frac{\left(\pi_n^{(1)}\right)^2}{\pi_n^{(0)}}. \tag{9}$$

One of the main properties of CFI is that $\mathcal{F}_C \leq \mathcal{F}$, since the QFI is the maximum of the CFI over all possible POVMs [69]. Therefore, $\mathcal{F}_C$ may potentially be zero while $\mathcal{F}$ is maximized. In this paper, we mainly focus on the maximization of QFI [16, 19–21]. The evolution of CFI is then obtained in a second step from the optimal controls derived for the QFI maximization and the selectivity problem. The computation of QFI can be difficult from Def. 1. Instead, we use another formulation, first derived in [76], and extended in [70].

**Theorem 1.** *Let $\rho_{X_0} = \rho^{(0)}$ be expressed in diagonal form, $\rho^{(0)} \equiv \sum_{k=0}^{s-1} p_k^{(0)}|\psi_k^{(0)}\rangle\langle\psi_k^{(0)}|$, where s is the dimension of the support of $\rho^{(0)}$, $p_k^{(0)} \in ]0,1]$ with $\sum_{k=0}^{s-1} p_k^{(0)} = 1$ and $\{|\psi_k^{(0)}\rangle\}$, $k = 0,\cdots,dim\ \mathcal{H}-1$, an orthonormal basis of $\mathcal{H}$. The Quantum Fisher Information is given by:*

$$\mathcal{F} = 4\sum_k \sum_{m|p_m^{(0)}>0} \frac{p_m^{(0)}}{(p_m^{(0)} + p_k^{(0)})^2} \left|\langle\psi_k^{(0)}|\rho^{(1)}|\psi_m^{(0)}\rangle\right|^2. \tag{10}$$

*For a pure state, $\mathcal{F}$ can be simplified into*

$$\mathcal{F} = 4\sum_{k>0}^{dim\mathcal{H}-1} \left|\langle\psi_k^{(0)}|\psi_0^{(1)}\rangle\right|^2, \tag{11}$$

*where $\rho = |\psi_0^{(0)}\rangle\langle\psi_0^{(0)}| + \delta X(|\psi_0^{(0)}\rangle\langle\psi_0^{(1)}| + |\psi_0^{(1)}\rangle\langle\psi_0^{(0)}|) + O(\delta X^2)$.*

Note that the ensemble of $|\psi_n^{(0)}\rangle$ is both a set of unperturbed quantum state and a basis of the Hilbert space.

**Theorem 2.** *The QFI of a pure state* (8) *can be written as*

$$\mathcal{F} = 4 \sum_{k>0}^{dim\mathcal{H}-1} \left| \langle \psi_0^{(0)} | A(t) | \psi_k^{(0)} \rangle \right|^2 , \tag{12}$$

*the operator $A(t)$ being defined as:*

$$A(t) = \left[ \int_0^t dt' \, U^{(0)\dagger}(t') \frac{\partial H(t')}{\partial X} U^{(0)}(t') \right]_{X=X0} , \tag{13}$$

*where $H$ is the Hamiltonian of the system, $U^{(0)}(t)$ the unperturbed evolution operator and $|\psi_0^{(0)}\rangle$ the initial state of the unperturbed system.*

The proof is given in Appendix A.2. Note that Eq. (12) and (11) are equivalent since $|\psi_0^{(1)}\rangle = A(t)|\psi_0^{(0)}\rangle$. Calculating the QFI explicitly can be tricky, and it may be easier to work with a tight upper bound [16]. This latter is used in this study to check the optimal character of the result and to describe the control protocol.

**Theorem 3.** *(Tight bound of the QFI)* . *For a pure state, we have*

$$\mathcal{F} \leq \left( \int_0^t dt_1 \, [\lambda_{\max}(t_1) - \lambda_{\min}(t_1)] \right)^2 , \tag{14}$$

*with $\lambda_{max}(t)$ and $\lambda_{min}(t)$ the maximum and minimum eigenvalues of $\partial_X H(t)$ at time $t$.*

A heuristic justification of this result was given in [16], and we provide a rigorous proof in Appendix A.3. The bound can be reached for a pure state which can be expressed as a linear combination with equal weights of the eigenstates associated with the maximum and minimum eigenvalues of $\partial_X H(t)$ at time $t$. Note that this state may not be generated by the dynamics if the physical system is not fully controllable [16].

Theorems 2 and 3 are useful for the design of control maximizing the QFI for a closed quantum system described by a pure state. If this is not the case, we have to consider Eq. (10), which can be difficult to handle in practice. An approximated expression of the QFI for mixed states is therefore interesting. We give below an original approximation of the QFI for a mixed state when it is close to a pure state.

**Theorem 4.** *(Approximated expression for the QFI of mixed states)*

*Let $\rho(t) = U(t, X) \left( |\psi_0\rangle\langle\psi_0| + \epsilon \sum_{k=0}^{dim\mathcal{H}-1} p_k^{(1)}(t)|\psi_k\rangle\langle\psi_k| \right) U^\dagger(t, X) + O(\epsilon^2)$ a density matrix for which $\epsilon$ and $p_k^{(1)}$ do not depend on the parameter $X$ to estimate. The QFI associated with this density matrix is given by:*

$$\mathcal{F} = 4 \sum_{n>0} |\langle\psi_0|A(t)|\psi_n\rangle|^2 (1 - \epsilon(p_0^{(1)} + 3p_n^{(1)})) + O(\epsilon^2), \tag{15}$$

*where $A(t)$ is defined by Eq.* (13).

The proof is given in Appendix A.4. This theorem is used in Sec. 4 to analyze the optimal strategy that allows us to maximize the QFI, and in Sec. 5.4 to compute the evolution of the QFI for specific examples. In the case of a two-level system where the eigenvalues of

$\partial_X H$ are such that $\lambda_{\min} = -\lambda_{\max}$, the result of Thm. 4 can be simplified by assuming that $|\psi_0\rangle$ is a linear combination of the two eigenvectors of $\partial_X H$ with equal weights. In this case, it is straightforward to show that $4\sum_{n>0} |\langle\psi_0|A(t)|\psi_n\rangle|^2$ has only one non-zero term, which leads to the maximum bound of Thm. 3. The QFI of the perturbed state is therefore $\mathcal{F} = \mathcal{F}|_{\epsilon=0}(1 - \epsilon(p_0^{(1)} + 3p_n^{(1)})) + O(\epsilon^2)$. We deduce from this formula that the relaxation process reduces the QFI, but this effect can be limited by a modulation of the coefficients $p_0^{(1)}$ and $p_1^{(1)}$.

# 3 Selective controls and their relation to quantum Fischer information

We introduce the notion of selective controls and we show to which extent this concept is connected to QFI. The underlying idea of selective control is to act differently on several systems characterized by different values of $X$. For our case, the density matrix $\rho_X$ is modified at the end of the control process only if $X = X_0$, and ideally, it is left unchanged otherwise. Due to the continuity of the density matrix with respect to $X$, and the discontinuous target states to be considered in the control problem, perturbation theory is not well suited to solve this issue. Instead, we adopt an alternative starting point by discretizing the parameter space [33, 34, 46, 48, 49].

**Definition 4.** *(Selectivity problem)* Let $\mathfrak{C} = \{X_0, X_1, ..., X_{N_s-1}\}$ be a set of $N_s$ values of the parameter $X$. A time-dependent density matrix $\rho_n$ is associated with each element $X_n$ of $\mathfrak{C}$. All these density matrices are simultaneously controlled with $u(t)$, $t \in [0, t_f]$. The goal of the control problem is to bring the systems with the same control $u(t)$ from $\rho_n(0) = \rho_{\text{ini}}$, $\forall\, 0 \geq n \geq N_s - 1$ to $\rho_n(t_f) = \rho_{\text{target},n}$ where $\{\rho_{\text{target},n}\}_{n\in N_s}$ is a set of target states associated with each $X_n$.

Many different control scenarios for the selectivity problem can be considered, but a standard one is to choose $\rho_{\text{target},0} = \rho_{\text{target}}$ and $\rho_{\text{target},n} = \rho_{\text{ini}}$ for $n \neq 0$ [33,34]. The ideal selective process which gives $\rho_X(t_f) = \rho_{\text{target}}$ if $X = X_0$ and $\rho_X(t_f) = \rho_{\text{ini}}$ otherwise is reached when the discretization grid approaches a continuum (i.e., the distance between nearby $X_n$ becomes infinitesimally small, and $N_S \to \infty$).

We now focus on a selectivity problem with $N_s = 2$ systems, and $\mathfrak{C} = \{X_0, X_0 + \delta X\}$, with $\delta X > 0$ kept fixed and finite. The QFI can be approached by a finite difference approximation, denoted $\mathcal{F}_{\text{fd}}$ and given at time $t_f$ by

$$\mathcal{F}_{\text{fd}}(t_f) = \frac{4}{(\delta X)^2} D(\rho_{X_0}(t_f), \rho_{X_0 + \delta X}(t_f))^2, \tag{16}$$

where the equality with the QFI is obtained when $\delta X \to 0$. $\mathcal{F}_{\text{fd}}(t_f)$ is maximized at time $t_f$ when $\text{Tr}\left[\sqrt{\sqrt{\rho_{\text{target}}}\rho_{\text{ini}}\sqrt{\rho_{\text{target}}}}\right] = 0$, or equivalently, when $D^2 = 2$ (see Eq. (6)). We deduce that $\max \mathcal{F}_{\text{fd}}(t_f) = 8/\delta X^2$, which also gives us the maximum value of $\mathcal{F}$ up to a term of order $\delta X$. We assume that there exists $\rho_{\text{target}}$ satisfying this property, and we denote $t_{\min}$ the minimum time to reach this state. For a closed quantum system, we have in many cases $t_{\min} \sim 1/\delta X$ for $\delta X$ small enough. We refer to [33,34] for examples on spin systems, but this property can be easily understood if $\delta X$ plays the role of an energy in the system Hamiltonian. In this case, the characteristic time of the associated process is typically of order $1/\delta X$. When $\delta X$ is not energy-like, this may not be the case, see [16] for a counter-example. For simplicity,

we assume below that $t_{\min} \sim 1/\delta X$ and we have

$$\mathcal{F} \simeq \frac{8}{\delta X^2} = 8\alpha t_{\min}^2, \tag{17}$$

with $\alpha$ a constant specific to the system. When $\delta X \to 0$, we obtain $t_{\min} \to \infty$ and $\mathcal{F} \to \infty$. This result shall be manipulated with caution because $\mathcal{F}_{\mathrm{fd}}$ gives us only an approximation of $\mathcal{F}$. However, in some cases, the optimization of the two quantities can lead to the same result.[1] This is the case when the optimization process which generates orthogonal states for a non-zero value of $\delta X$ coincides with the optimal control of the QFI. Several examples of such a situation are given in Sec. 5. Notice that orthogonal states may be generated only in infinite time for two infinitesimally close parameters. We shall also stress that both $\alpha$ and $t_{\min}$ depend on the target state, and thus the choice of this latter has a consequence on the control duration. The global maximum is obtained when $\rho_{\mathrm{target}}$ is chosen so that the corresponding minimum time is the smallest among all possible minimum times associated with all the target states that verify $D^2 = 2$.

Nevertheless, the solution may not be unique, and in the case when the target state may not be reached exactly, we have no guarantee that the solution to the two problems is equivalent. In Sec. 5, we explore, by means of an example, under which conditions the two approaches lead to the same control mechanism. As already discussed, experimentally it may be more relevant to consider CFI rather than QFI. The selectivity problem can be connected to the maximization of CFI if $\rho_{\mathrm{ini}}$ and $\rho_{\mathrm{target}}$ are taken such that a maximum of sensitivity is achieved in the measurement basis. In this case, selective controls may not lead to the maximum QFI. This point is illustrated in Fig. 2, and it is discussed in Sec. 5 with a case study.

## 4 Methodology for optimal control design

Optimality criteria are rigorously defined within the framework of optimal control theory [56–58, 77]. The main idea is to define a cost function (or figure of merit) that encodes quantitatively the objective(s) of the control process. This function is then minimized (or maximized) with respect to the control parameter. In this study, we consider two types of cost functions to minimize:

$$C = -\mathcal{F}(t_f), \tag{18}$$

for the maximization of QFI, and

$$C = \frac{1}{N_s} \sum_{n=1}^{N_s} D(\rho_{\mathrm{target},n}, \rho_n(t_f))^2 + t_f, \tag{19}$$

for the design of a time-optimal selective control. In the case of QFI, the cost function aims at reaching the largest value of QFI at time $t_f$. For the selectivity problem, the cost has a double objective since the $N_s$ systems must reach their respective target states as fast as possible (hence the term $t_f$ in Eq. (19)). The minimum time constraint is motivated by the fact that if

---

[1]This statement can be justified qualitatively from a Taylor expansion of the function and an exact treatment of the reminder. Consider for instance a function $f$ such that $f(0) = df/dx(0) = 0$ (this is the case for the QFI). Then, $\exists c \in ]0, b[ \mid d^2f/dx^2(c) = f(b)2b^2$. Note the strict equality between the second derivative of the function at $x = c$ and the function itself at $x = b$. Here $d^2f/dx^2(0)$ and $f(b)$ play respectively the role of $\mathcal{F}$ and $\mathcal{F}_{fd}$. For a fixed value of $b$, chosen small enough, we deduce that maximizing $f(b)$ also amounts to maximizing $d^2f/dx^2(c)$. If the variations of the second derivative $d^2f/dx^2$ are not too strong in the small interval $[0, c]$, this also amount to maximizing $d^2f/dx^2(c)$. The same kind of argument can be used for the QFI. This analysis in terms of Taylor expansion is not discussed in depth here because it requires careful treatment of the logarithmic derivative operator $L$ (to be mathematically well justified).

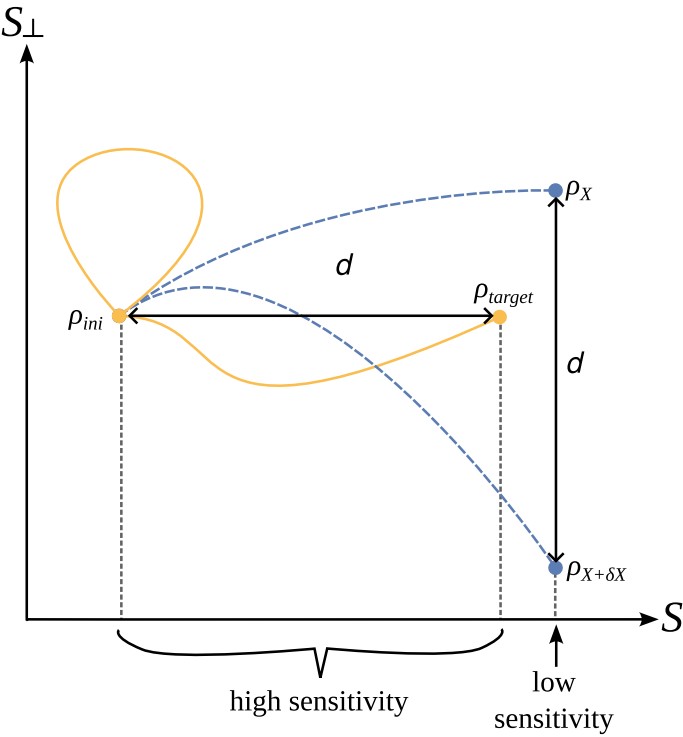

Figure 2: Schematic illustration of the difference between time-optimal selectivity problem and optimization of the QFI in the case when the target states are taken to maximize the CFI. The plane $(S, S_\perp)$ represents the space of density matrices. The axis $S$ depicts the subspace of density matrices which can be determined by the POVM associated with a given experimental setup, and $S_\perp$ is the subspace of density matrices that return the same measurement results. The initial state of the system is given by the point $\rho_{\text{ini}}$. The trajectories of two systems associated with different values of $X$ are given by yellow and blue-dashed lines, depending on whether the control is optimized for the selectivity problem or the QFI. The final distance is chosen equal for the two problems, but the duration of each process can be different. In the case of QFI, we observe that the sensitivity of the measurement can be very low if the two states are in $S_\perp$, while for the selectivity problem, we have a very good precision since the two states are in $S$.

the distance between the target states is maximum then there is a minimum time for such a transformation (see Sec. 3), and for well-chosen target states, this corresponds to the solution of QFI maximization.

Different methods can be used for this purpose. Gradient-based algorithms such as GRAPE (Gradient Ascent Pulse Engineering) [78] are particularly efficient, but other methods based on the Pontryagin Maximum Principle (PMP), such as shooting algorithms, are another option [19, 49, 57]. The technical aspects of the design of time-optimal selective controls with the PMP have been considered in [33, 34], and we refer the interested reader to these articles for details.

We now focus on the control strategy to use for maximizing QFI. We first consider pure states and then we extend the discussion to mixed states. From Thm. 2 and 3 , we observe that the upper bound can be reached if in Eq. (12), $\langle \psi_0^{(0)}(t)|\partial_X H(t)|\psi_k^{(0)}(t)\rangle$ is maximized at any time $t$. If the vectors $|\psi_k(0)\rangle$ are eigenvectors of $\partial_X H$ then $\langle \psi_0^{(0)}(t)|\partial_X H(t)|\psi_k^{(0)}(t)\rangle$ is maximized when $|\psi_0^{(0)}\rangle$ is a superposition of the eigenvectors associated with the smallest and

highest eigenvalues, and hence, we recover the bound of Thm. 3. When the system is sufficiently controllable, the optimal trajectory can be determined from the following procedure:

1. Generate with a specific control, a state which maximizes $\langle\psi_0^{(0)}(t_1)|\partial_X H(t)|\psi_k^{(0)}(t_1)\rangle$, where $t_1$ is the control duration of this first control process. The choice of the state may be non-unique.

2. Use a control to stabilize the system, i.e., such that for all $t \geq t_1$, $\langle\psi_0^{(0)}(t)|\partial_X H(t)|\psi_k^{(0)}(t)\rangle$ remains on its maximum value.

Note that this control strategy has already been discussed in [16, 19, 20]. We push the analysis further by expanding the QFI into a sum of contributions associated with each part of the control process. Using Thm. 2, we have:

$$
\begin{aligned}
C &= -4 \sum_{k>0}^{dim\mathcal{H}-1} \left| \int_0^{t_f} dt_1 \langle\psi_0^{(0)}(t_1)| \frac{\partial H(t_1)}{\partial X} |\psi_k^{(0)}(t_1)\rangle \right|^2 \\
&= -4 \sum_{k>0}^{dim\mathcal{H}-1} \left| \langle\psi_0^{(0)}(0)|A(0,t_1)+A(t_1,t_f)|\psi_k^{(0)}(0)\rangle \right|^2 ,
\end{aligned}
\tag{20}
$$

where $A(t_a,t_b)$ corresponds to the integral between $t_a$ and $t_b$ of $U^{(0)^\dagger}\partial_X H U^{(0)}$. The integral is split into two parts, to separate the initialization and stabilization processes. The cost $C$ can be written as follows:

$$
C = -\mathcal{F}_1(0,t_1) - \mathcal{F}_2(t_1,t_f) - \mathcal{F}_{12}(0,t_1,t_f),
\tag{21}
$$

where $\mathcal{F}_1(0,t_1)$ is the QFI associated with the initialization process, $\mathcal{F}_2(t_1,t_f)$ the one corresponding to the stabilization protocol, and

$$
\mathcal{F}_{12}(0,t_1,t_2) = 8 \sum_{n>0} \Re\left[ \langle\psi_0^{(0)}(0)|A(0,t_1)|\psi_k^{(0)}(0)\rangle \overline{\langle\psi_0^{(0)}(0)|A(t_1,t_f)|\psi_k^{(0)}(0)\rangle} \right],
\tag{22}
$$

a cross-term between the two parts of the process.

Since the stabilization procedure maximizes the increase of QFI at any time, $C$ is bounded from below by $C_{\min} = -\mathcal{F}_2(0,t_f)$. This minimum is reached in the limit when the initialization process is infinitely short, i.e. when the time $t_1$ goes to 0. Therefore, the optimal strategy is given by a time-optimal state-to-state transfer from the initial state to the state maximizing the QFI. If there are several equivalent time optimal solutions, we select the one(s) that maximizes $\mathcal{F}_1 + \mathcal{F}_{12}$. Finally, we discuss the case of open quantum systems with the QFI given by the approximated expression of Thm. 4. The environment is responsible for a QFI change of the order of $\epsilon$. Depending on the coefficients $p_n^{(1)}$, QFI can be modulated by using the environment as a resource. A specific choice of states or control could allow us to reduce the detrimental relaxation effect. This point is not trivial and system dependent. It is discussed in Sec. 5 for the case study of a spin-$\frac{1}{2}$ particle.

# 5 Example of a spin system coupled to a bosonic bath

As a concrete example, we study the estimation of the values of the Hamiltonian parameters of a spin-$\frac{1}{2}$ system coupled to a bosonic reservoir at zero temperature. We first define the model system. We then compute the optimal controls of the selectivity problem and the ones maximizing the QFI. We then show to which extent the system parameters can be estimated.

A comparison between the different controls is performed using the Bures distance, the QFI, and the CFI. Finally, we explore how two of the optimized controls behave in a simulated experiment in which the spin frequency is found using the maximum likelihood method. We also discuss the local/non-local properties of the methods.

## 5.1 The model system

We consider a spin-$\frac{1}{2}$ system whose Hilbert space is $\mathcal{H} = \mathbb{C}^2$. In the case of a weak Jaynes-Cummings interaction and a Lorentzian spectral density for the bath at zero temperature, and under a Markovian approximation, the state of the spin is described by a density matrix $\rho_S$ whose dynamics are governed by the Lindblad equation (in $\hbar$ units) [66]:

$$\frac{d\rho_S}{dt} = -i[H(t), \rho_S(t)] + \gamma \, D_{\sigma_+}[\rho_S(t)], \tag{23}$$

with

$$H(t) = -\frac{\Delta}{2}\sigma_z - \frac{\alpha}{4}\omega(t)\left(e^{-i\phi(t)}\sigma_+ + e^{i\phi(t)}\sigma_-\right), \tag{24}$$

$$D_{\sigma_+}[\rho_S(t)] = \sigma_+\rho_S(t)\sigma_- - \frac{1}{2}\left(\sigma_-\sigma_+\rho_S(t) + \rho_S(t)\sigma_-\sigma_+\right), \tag{25}$$

where $\sigma_z$, $\sigma_x = \sigma_- + \sigma_+$, $\sigma_y = i(\sigma_- - \sigma_+)$ are Pauli matrices, $\Delta$ is the detuning (also called offset) of the spin frequency with respect to a reference frequency $\omega_S$, $\omega(t) \in [0, \omega_0]$ and $\phi(t) \in ]-\pi, \pi]$ are two bounded controls, $\alpha$ is a scaling factor for the amplitude of the control, and $\gamma$ gives the relaxation rate of the spin energy into the bath. We are interested in the estimation of the parameters $\Delta, \alpha$, and $\gamma$ which are assumed to be constant. For this case study, we choose the exact values to be $\Delta_0 = 0.2\,\omega_0$, $\alpha_0 = 1$, and $\gamma_0 = 0.05\,\omega_0$, which are compatible with magnetic resonance experiments [33,46,49]. The ratio $\Delta/\gamma$ is also compatible with ultra-cold spin systems, since the offset is usually in the MHz range, and the relaxation rate is in the kHz range [79].

For this example, the POVM is given by the two projectors associated with the eigenstates of $\sigma_z$. The states are denoted by $|\uparrow\rangle$ and $|\downarrow\rangle$ (associated respectively with the eigenvalues $+1$ and $-1$ of $\sigma_z$). Note that this choice is motivated by the aim of providing simple comparisons between the different approaches. The initial state of the system is the steady state of the relaxation operator, $\rho_{\text{ini}} = |\uparrow\rangle\langle\uparrow|$.

As underlined in Sec. 4, the control protocols can be found using numerical optimization methods. For this case study, we already have a precise characterization of the time-optimal synthesis [33,80,81] in terms of piecewise constant pulses. It is therefore natural to solve this control problem by using this control family in which the control amplitude, its phase, and the duration of a time step are the parameters to find. In the free relaxation case, the time minimum selective solution for the spin frequency is the concatenation of three different constant controls. On this basis, we limit here our search to a family with five time-steps, which leads to a good compromise between computational time and control efficiency. The values of the 15 parameters to estimate in a bounded domain can be obtained with a standard global optimization algorithm, such as the simulated annealing algorithm of Mathematica [82]. In the minimization of the cost (18), the final time is fixed, while it is free for the figure of merit (19). In this latter case, the optimization is performed for different values of the final time $t_f$, in order to find the solution minimizing simultaneously the cost and the control duration. In a specific case (estimation of the relaxation rate), we have checked the optimality of the results with a shooting algorithm based on the PMP [49].

## 5.2 Estimation of the offset frequency Δ

The estimation of the parameter $\Delta$ is performed in two steps. First, we find the optimal control strategy without relaxation (i.e., $\gamma = 0$), and then, we discuss how the strategy can be adapted to limit the effect of the environment when $\gamma \neq 0$. In the numerical simulations, we use the following values for the other parameters, $\Delta_0 = 0.2 \, \omega_0$ and $\alpha_0 = 1$.

The optimal procedure that maximizes $\mathcal{F}$ in a free-relaxation case has already been investigated in [16]. We briefly recall this solution below. For a pure state, the QFI associated with the estimation of $\Delta$ is

$$\mathcal{F} = \sum_{k>0}^{\dim\mathcal{H}-1} \left| \int_0^{t_f} dt_1 \langle \psi_0^{(0)}(0)|U_S^{(0)^\dagger}(t_1) \, \sigma_z \, U_S^{(0)}(t_1)|\psi_k^{(0)}(0)\rangle \right|^2 . \tag{26}$$

Following the procedure described in Sec. 4, the goal is to stabilize the system on a specific state, which maximizes the growth rate of Eq. (26). Using Thm. 3, it is straightforward to find this state which corresponds to a state superposition of $|\uparrow\rangle$ and $|\downarrow\rangle$ with equal population. This state is located on the equator of the Bloch sphere (i.e. $|\psi_0^{(0)}\rangle = \frac{1}{\sqrt{2}}(|\uparrow\rangle + e^{i\theta}|\downarrow\rangle)$) with $\theta \in [0, 2\pi)$). For $\Delta$ small enough (i.e. $\Delta \leq 0.5\omega_0$) [46,47], the time optimal solution to reach the equator is given by a control of maximum amplitude with $\omega(t) = \omega_0$, $\phi(t)$ is an arbitrary constant and the pulse duration can be expressed as $t_{\text{pulse}} = 4\arcsin\left(\sqrt{\omega_0^2 + 4\Delta_0^2}/(\omega_0\sqrt{2})\right)/\sqrt{\omega_0^2 + 4\Delta_0^2}$. Once the optimal state is reached, the growth rate of $\mathcal{F}$ must be stabilized. This is obtained when the term $U_S^{(0)^\dagger}(t_1)\sigma_z U_S^{(0)}(t_1)$ in Eq. (26) does not depend on time, i.e. $\omega(t_1) = 0$ for all times $t_1$ since this condition reduces the Hamiltonian to $H = -\Delta\sigma_z/2$. It is then straightforward to see that the different evolution operators commute with $\sigma_z$ and they cancel each other. Another option consists in using $\phi(t_1) = \Omega t_1$ with $\Omega \gg \Delta$. This produces an effective Hamiltonian with a coupling constant $\alpha/\Omega$ which is negligible when $\Omega \to \infty$ (see [35] for recent discussions on this effect).

We compare this optimal strategy with the one of the selectivity problem where $\mathfrak{C} = \{-\Delta_0, \Delta_0\}$. The target states associated with these offsets are respectively given by $\rho_{\text{target},-\Delta_0} = |\uparrow\rangle\langle\uparrow| = \rho_{\text{ini}}$, and $\rho_{\text{target},\Delta_0} = |\downarrow\rangle\langle\downarrow|$. Without relaxation, the time optimal solution has been derived in [33]. The control is the concatenation of three constant pulses. The optimal solution is plotted in Fig. 3 (note that this control is not unique). The control mechanism can be described as follows. The first pulse transfers the initial state (north pole of the Bloch sphere) to the equator of the Bloch sphere. Then, the second part, with a zero amplitude, induces a free evolution of the spins along the equator. Due to the detuning difference, they rotate in opposite directions. The duration of this process is chosen so that with the application of the third pulse, the spin with $\Delta = \Delta_0$ reaches the south pole while the other spin goes in the opposite direction and returns to the north pole. Note that the optimal control mechanism used for this selectivity problem is the same as the optimal control procedure for the QFI maximization, except for the last control part.

In Fig. 3, we compare the different controls by plotting the Bures distance (between the states associated with $-\Delta_0$ and $\Delta_0$), the QFI and the CFI as a function of time. As expected, the two controls lead to similar evolutions until the very end of the control process. In the two cases, the Bures distance can reach the maximum value of 2. This value is achieved first by the optimal control of the QFI. This is because the two spins can have orthogonal states when they are both on the equator of the Bloch sphere. This configuration is easier to achieve from the initial state since we have to supply a smaller amount of energy to the system. Concerning the QFI, we notice that the two control strategies are quite close to the upper bound, the difference being mainly due to the non-zero duration of the control used to generate a state on the Bloch sphere (we recall that the equator is the set of states maximizing the increase of QFI). The

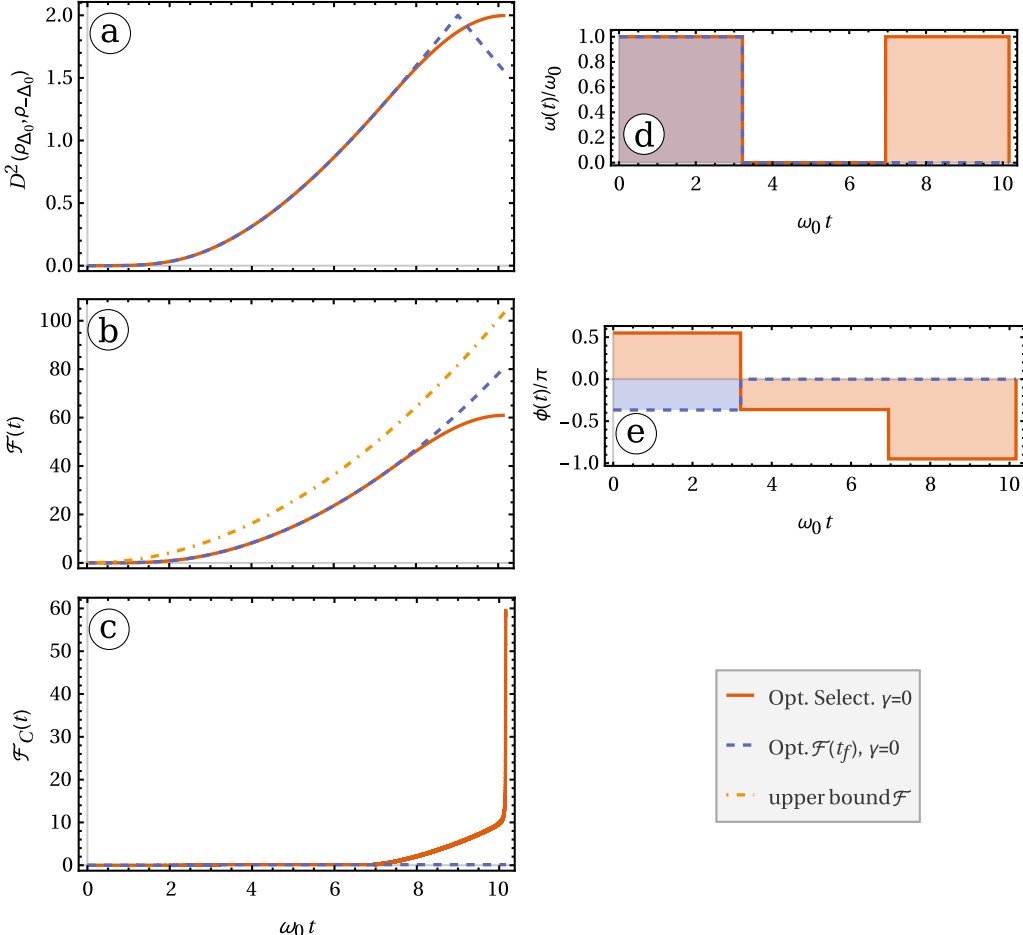

Figure 3: a) Time evolution of the Bures distance between the density matrices of two spins associated with the offsets $-\Delta_0$ and $\Delta_0$ when the two spins are simultaneously driven by the optimal selective control or the one maximizing the QFI. b) QFI for a spin of offset $\Delta_0$ when it is driven by such controls. The maximum bound of the QFI, calculated with Thm. 3, is also indicated. c) Same as panel b), but with CFI instead of QFI. Panels d) and e) show respectively $\omega(t)$ and $\phi(t)$ for the two controls used in the simulations. The final time $t_f = 3.235\pi/\omega_0$ is given by the duration of the time optimal selective control. Numerical parameters are set to $\Delta_0 = 0.2\,\omega_0$, $\alpha_0 = 1$, and $\gamma_0 = 0$.

optimal selective control has a slightly lower final QFI value because the spin state must leave the equator of the Bloch sphere. Finally, the CFI is the quantity with the most important difference. Here, the optimal control of the QFI leads to a constantly zero CFI, which is due to the orthogonality of the optimal measure basis for $\Delta$ and the basis $\{|\uparrow\rangle, |\downarrow\rangle\}$. However, with the optimal selective control, we reach the maximum value of the CFI at the very end of the control process, because the target states correspond to the measurement basis. In this situation, we have been able to determine numerically that the optimal control maximizing the CFI is the one given by the selective control. The optimization is performed in the same way as maximizing the QFI, but with CFI as the terminal cost. This is a difficult task because the optimization algorithm is easily trapped in a local maximum of the function.

Next, we study the role of the environment in the estimation of $\Delta$ with $\gamma = \gamma_0 = 0.05\,\omega_0$. Results similar to those in Fig. 3 are shown in Fig. 4 for the relaxation case. We observe that the maximum values of the Bures distance and the QFI are significantly smaller than the

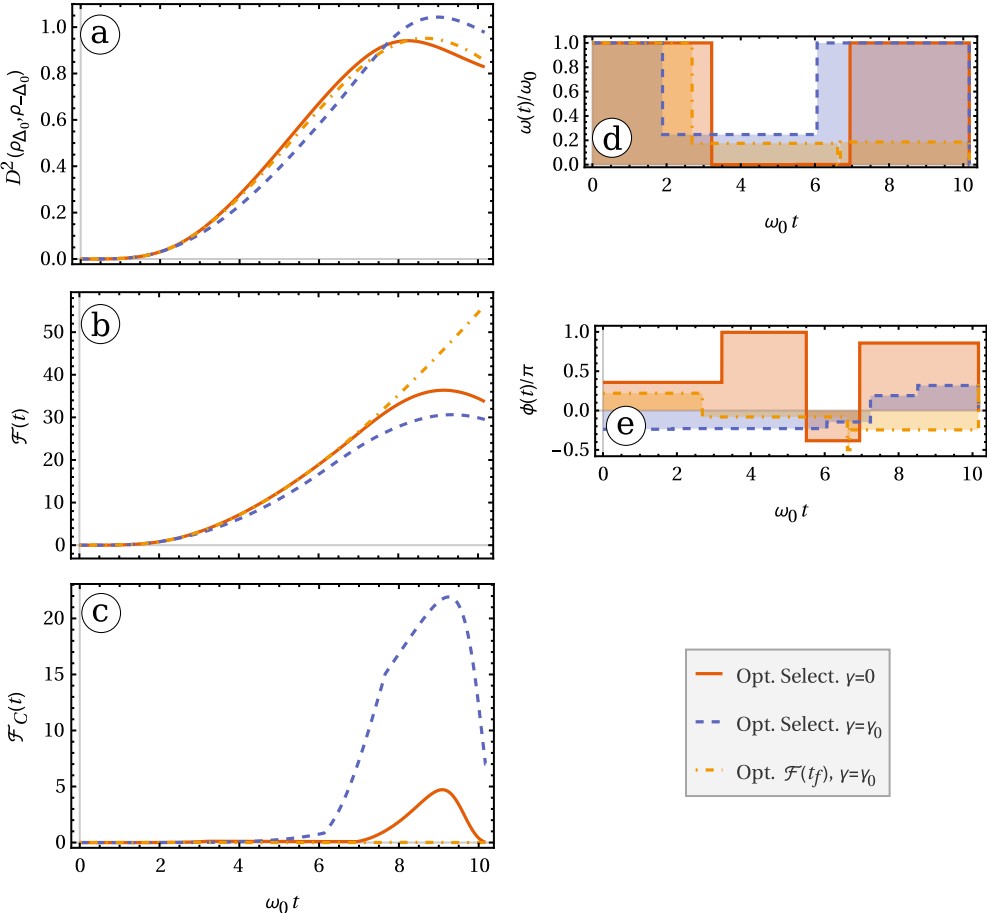

Figure 4: a) Time evolution of the Bures distance between the density matrices of two spins associated with the offsets $-\Delta_0$ and $\Delta_0$ when the two spins are simultaneously driven by the optimal selective control computed with $\gamma = 0$, the one with $\gamma = \gamma_0$, and the optimal control for the QFI optimized with $\gamma = \gamma_0$. b) Same as panel a), but for the QFI of a spin with $\Delta = \Delta_0$. c) Same as panel a), but for the CFI of a spin with $\Delta = \Delta_0$. Panels d) and e) depict respectively $\omega(t)$ and $\phi(t)$. The final time is fixed to $t_f = 3.235\pi/\omega_0$. The other parameters are set to $\Delta_0 = 0.2\ \omega_0$, $\alpha_0 = 1$, and $\gamma_0 = 0.05\ \omega_0$.

ones obtained when $\gamma = 0$ (the difference is larger than a factor 2). In Fig. 4, we consider three different controls, i.e. (i) the optimal selective solution computed with $\gamma = 0$, (ii) the optimal selective one derived with $\gamma = \gamma_0$, and (iii) the optimal control for the QFI optimized with $\gamma = \gamma_0$. We note the behaviors opposite to those of Fig. 3. Here, a link between the Bures distance and the QFI is no longer observed. For example, maximizing the selectivity leads to the largest Bures distance, but to the smallest QFI, even for the optimal solution with $\gamma = 0$. Moreover, maximizing the QFI does not lead to a significant increase in the Bures distance. Since we cannot reach the largest values of $\mathcal{F}$ and $D^2$, we cannot easily relate the QFI maximization and selectivity problem, and corrections to the optimal strategies due to relaxation effects are specific to control problems. The maximum values of $\mathcal{F}$ and $D^2$ are significantly larger for controls optimized with $\gamma = \gamma_0$ than for $\gamma = 0$. However, in the case of Bures Distance, the maximum is reached before the final time. This means that the control duration can be reduced to obtain better measurement sensitivity. Concerning the CFI, again, we have constantly zero with the optimal control of the QFI, and larger values with selective controls. The best CFI is achieved from the protocol optimized with $\gamma = \gamma_0$.

## 5.3 Estimation of the relaxation rate $\gamma$

For the estimation of the parameter $\gamma$, an approximated expression of the QFI can be derived by assuming that $\gamma_0 = 0$. In this situation, a first-order time-dependent perturbation theory in powers of $\gamma$ leads to:

$$\rho_S(t) = U_S(t)\left(\rho_S(0) + \gamma t\left\{\sigma_+\rho_S(0)\sigma_- - \frac{1}{2}(\sigma_-\sigma_+\rho_S(0) + \rho_S(0)\sigma_-\sigma_+)\right\}\right)U_S^\dagger(t) + O(\gamma^2),$$
(27)

with $U_S(t) = \mathbb{T}\exp\left(-i\int_0^t dt' H(t')\right)$. Then it is straightforward to apply Thm. 1, since $\rho_S^{(0)}$ is a pure state, and $\rho_S^{(1)}$ is essentially given by $D_{\sigma_+}[\rho_S^{(0)}(0)]$. We obtain:

$$\begin{aligned}
\mathcal{F} &= t^2\left|\langle\psi_0^{(0)}(0)|D_{\sigma_+}\left[|\psi_0^{(0)}(0)\rangle\langle\psi_0^{(0)}(0)|\right]|\psi_0^{(0)}(0)\rangle\right|^2 \\
&= t^2\left|\langle\sigma_-\rangle\langle\sigma_+\rangle - \langle\sigma_-\sigma_+\rangle\right|^2 \\
&= t^2\left|\langle\downarrow|\psi_0^{(0)}(0)\rangle\right|^8.
\end{aligned}$$
(28)

The initial state must be $|\downarrow\rangle$ in order to obtain the maximum increase of $\mathcal{F}$ as a function of time. However, this procedure cannot be performed when $\gamma_0 > 0$ since the south pole of the sphere cannot be reached exactly. To illustrate this point, we set[2] $\Delta_0 = 0$. We also use $\gamma_0 = 0.05\omega_0$ and $\alpha_0 = 1$. Numerical results are given in Fig. 5. We observe that the optimal control that maximizes the QFI is given by a square pulse bringing the spin to a state such that $z \simeq -0.8$, followed by a free relaxation. The end of the constant pulse corresponds to the time when the $z$-coordinate is minimum (it is the smallest reachable value). Then, the optimal solution is the closest trajectory to the one given in Eq. (28).

A selectivity problem can also be defined for the estimation of $\gamma$. We set $\mathfrak{C} = \{\gamma_0, \gamma_0 + \delta\gamma\}$, with $\delta\gamma$ of the order of $\gamma_0$ (in the simulations, $\delta\gamma = \gamma_0$). Contrary to the estimation of $\Delta$, the center of the Bloch sphere is chosen as the target state of the first spin (i.e. $\rho_{\text{target},1} = \mathbb{I}/2$), and the initial state as the target state for the second spin. This choice is motivated by the fact that, in this case, the two poles of the sphere cannot be reached simultaneously. This is due to the relaxation process that induces dynamics inside the Bloch ball, most of the points of the Bloch sphere are therefore not accessible. Moreover, in this studied example, the target states cannot be attained exactly (due to the non-unitarity of the dynamics), and only states leading qualitatively to the desired behavior can be chosen. The initial state is the attractor of the relaxation process, it is therefore an interesting target state which can be generated quite easily. The center of the Bloch ball is less easily reached but it is inside the ball, and this allows us to capture a property similar to the south pole of the Bloch sphere. We stress that these target states are not orthogonal, but they avoid the search for suboptimal solutions with respect to the goal of increasing the distance between the two systems.

We observe that the optimal solution is equivalent to the one maximizing the QFI. This agreement between the two approaches allows us to interpret physically the optimal control of the QFI. The control steers the system to a state for which a variation of $\gamma$ (with respect to the reference $\gamma_0$) produces a maximum difference along the $z$-axis of the Bloch sphere. In particular, when a spin with $\gamma = \gamma_0$ is at $z = 0$, a spin with $\gamma > \gamma_0$ is as close as possible to the ground state.

We end this section with some comments on the generalization of the selectivity problem for more general couplings with the environment. If we introduce damping operators $D_{\sigma_-}$ and

---

[2]This can be done if we already know the spin energy transition perfectly, and if we use a frame rotating at this frequency.

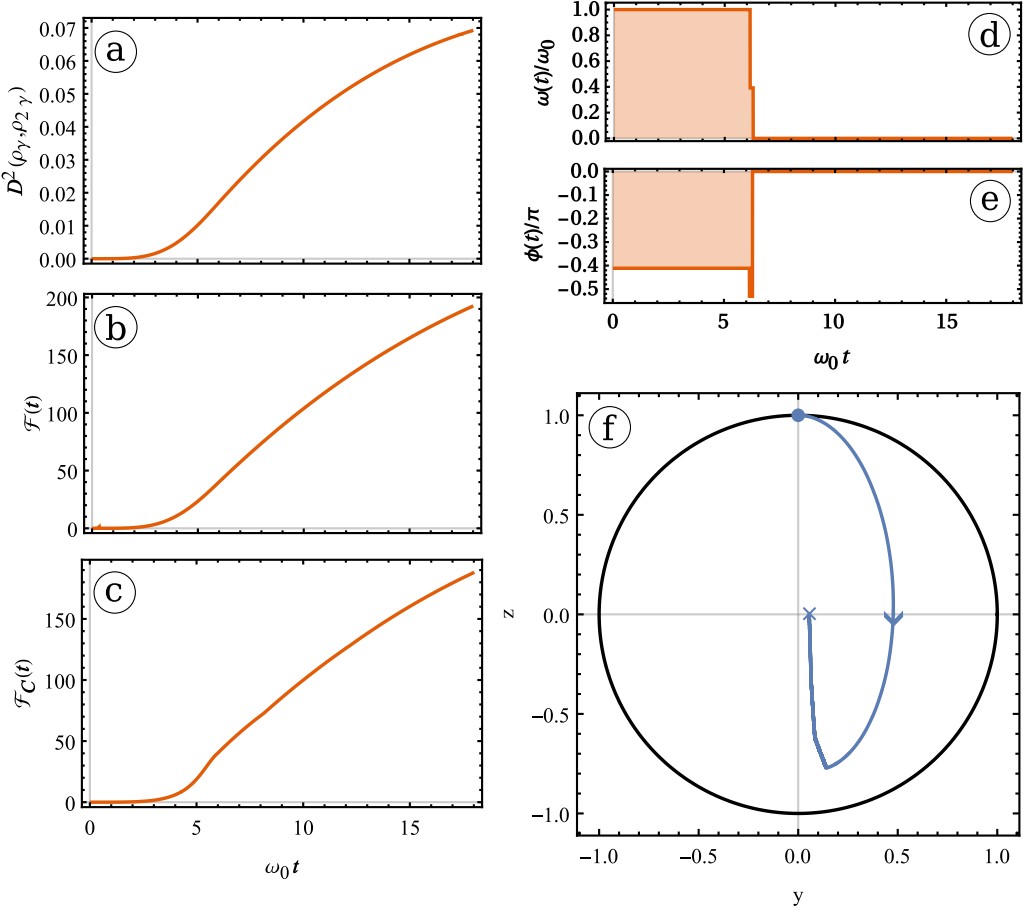

Figure 5: a) Time evolution of the Bures distance between the density matrices of two spins associated with the rates $\gamma_0$ and $2\gamma_0$ when the two spins are simultaneously driven by the optimal control maximizing the QFI. b) Same as panel a), but for the QFI of a spin with $\gamma = \gamma_0$. c) Same as panel a), but for the CFI. Panels d) and e) show respectively $\omega(t)$ and $\phi(t)$. Panel f) depicts the trajectory of the spin with $\gamma = \gamma_0$ in a slice of the Bloch ball (the initial state of the trajectory is given by a point and the final state by a cross). The final time $t_f$ is given by the duration required to reach approximately the center of the Bloch ball. Numerical parameters are set to $\Delta_0 = 0$, $\alpha_0 = 1$, and $\gamma_0 = 0.05\,\omega_0$.

$D_{\sigma_z}$ in the Lindblad equation, this latter is significantly modified [83], and the selective control strategy presented in this paragraph is, in general, no longer optimal. Moreover, with the use of the operators $D_{\sigma_-}$ and $D_{\sigma_z}$, the selectivity problem becomes very similar to the contrast problem of spin-$\frac{1}{2}$ particles in Nuclear Magnetic Resonance, which has been already studied extensively (see [36–39] for details).

## 5.4 Estimation of the control scaling factor $\alpha$

As a third example, we investigate the estimation of the parameter $\alpha$ with $\alpha \simeq \alpha_0 = 1$ by starting the analysis of the QFI. In the case $\gamma = 0$, the state is pure, and Eq. (12) can be used to compute $\mathcal{F}$. The core of the computation is the derivation of $A(t)$, defined in Eq. (13). Using a first-order expansion of $U^{(0)}(t)$ in power of $\alpha_0$, we obtain:

$$A(t) = \int_0^t dt_1 \left( \frac{\omega(t_1)}{4} \left( e^{i(-\phi(t_1)+\Delta t_1)}\sigma_+ + e^{-i(\phi(t_1)-\Delta t_1)}\sigma_- \right) + 4i\alpha_0 v(t_1)\sigma_z \right) + O(\alpha_0^2), \quad (29)$$

with

$$v(t_1) = \frac{1}{16} \int_0^{t_1} dt_2 \, \omega(t_1)\omega(t_2) \cos\left(\Delta(t_1 - t_2) - \phi(t_1) + \phi(t_2)\right). \tag{30}$$

We point out that if $|\psi_k^{(0)}\rangle$ is an eigenvector of $\sigma_z$, the term in $\sigma_z$ is canceled and the QFI does not depend on $\alpha_0$. Moreover, for the case $\phi(t) = \Delta t$, $\omega(t) = \omega_0$, the QFI becomes:

$$\mathcal{F} = \omega_0^2 t^2. \tag{31}$$

This is the upper bound, which can be calculated with Eq. (14). Interestingly, this is the same solution as the one that can be obtained with $\alpha_0 = 0$. Since the initial state can be chosen equal to $|\uparrow\rangle$, we do not need to prepare the system as in the case of Sec. 5.2. We can verify that this control is also the optimal solution for a selectivity problem with $\mathfrak{C} = \{\alpha_1, \alpha_2\}$, and the target states $\rho_{\text{target},\alpha_1} = |\uparrow\rangle\langle\uparrow| = \rho_{\text{ini}}$, and $\rho_{\text{target},\alpha_2} = |\downarrow\rangle\langle\downarrow|$. Next, we study the role of the relaxation effect on the QFI if the initial state is either $|\uparrow\rangle\langle\uparrow|$ or $|\downarrow\rangle\langle\downarrow|$. No simple expression can be derived in this case, but the result of Thm 4 can be used. We have respectively:

$$\mathcal{F}_{|\uparrow\rangle\langle\uparrow|} = \mathcal{F}|_{\gamma=0}, \tag{32}$$

$$\mathcal{F}_{|\downarrow\rangle\langle\downarrow|} = \mathcal{F}|_{\gamma=0}(1 - 2\gamma t), \tag{33}$$

where $\mathcal{F}|_{\gamma=0}$ is the QFI computed with $\gamma = 0$, as the one in Eq. (31). This suggests that this solution cannot be improved from an additional control. Our numerical investigation leads to the same conclusion, both for QFI and selectivity.

## 5.5 Comparison of the methods with a maximum likelihood estimation and the role of non-local effects

In the previous sections, we have compared the different methods of estimating several parameters of a spin-$\frac{1}{2}$ system, using Bures distance, QFI, and CFI as quantifiers of control efficiency. However, these quantities do not exactly describe the estimation process. In this section, we approach a real experiment, and we simulate the estimation of the parameter $\Delta$ with the two optimized controls given in Fig. 3. This analysis leads us to additional comments on the local/non-local properties of the different methods.

We assume that we have a first estimate of $\Delta$, given by $\Delta = 0.2 \pm 0.4 \, \omega_0$, and the other parameters are known perfectly, $\alpha = 1$ and $\gamma = 0$. The knowledge that we have on $\Delta$ is compatible with the two controls represented in Fig. 3, with $\Delta_0 = 0.2 \, \omega_0$. To simulate the fact that $\Delta_0$, used in the computation of the control, may not be the correct value, we introduce the "true" offset frequency $\Delta_\star = 0.25 \, \omega_0$, which must be found by the estimation process. The simulated experimental data associated with each control are made of 50.000 independent measurements of $\sigma_z$, randomly sampled with the probability distribution given by $P_\uparrow = |\langle\uparrow|\rho(t_f, \Delta_\star)|\uparrow\rangle|^2$ and $P_\downarrow = |\langle\downarrow|\rho(t_f, \Delta_\star)|\downarrow\rangle|^2$ (we recall that $\sigma_z$ is the measurement operator that allows us to define the POVM used to compute the CFI in the previous sections). From the two simulated data, the parameter $\Delta$ is determined using the maximization of the loglikelihood function, and the uncertainty is computed with a bootstrap of the data [62, 63]. The bootstrap method works as follows. From the initial series of measurements, new series are created by resampling the data. For each resampled data, we estimate the value of $\Delta$ with the maximum likelihood method. For each case, we obtain a different value of the parameter, leading to a mean value and an uncertainty.

In Fig. 6 we plot the normalized histograms of values of $\Delta$ obtained from the two simulated experiments. They represent the probability density $P_{\text{bootstrap}}(\Delta)$ of reaching a value of $\Delta$ with the maximization of the log-likelihood function. Using the control maximizing the QFI, two well-localized peaks are observed. This is because, in this case, $P_\uparrow = |\langle\uparrow|\rho(t_f, \Delta)|\uparrow\rangle|^2$ and

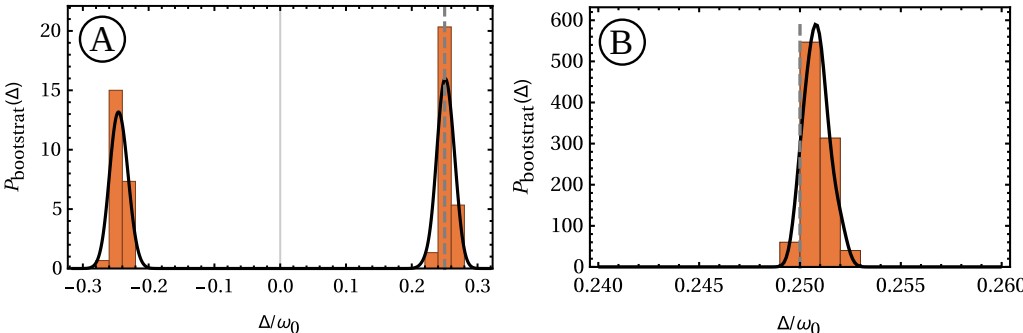

Figure 6: Normalized histogram of values of $\Delta$ determined by the log-likelihood method, for the control maximizing the QFI (panel A), and for an optimal selective control (panel B). The controls correspond to the two ones shown in Fig. 3. The data used for the estimation of $\Delta$ is a numerical simulation of experimental measurements (see the main text for details). The normalization coefficients of the histograms is such that $\int P_{\text{bootstrap}}(\Delta) \, d\Delta = 1$. The black curves are smoothed versions of the histograms. The vertical dashed lines show the value of $\Delta_\star$ that must be recovered by the estimation process.

$P_\downarrow = |\langle \downarrow |\rho(t_f, \Delta)| \downarrow \rangle|^2$ are identical for $\pm\Delta$. With optimal selectivity, the symmetry is broken since negative and positive offsets are moved to different hemispheres of the Bloch sphere. For each peak independently, we have: $\Delta = -0.245 \pm 0.016 \, \omega_0$ and $\Delta = 0.251 \pm 0.015 \, \omega_0$ for the QFI, and $\Delta = 0.25095 \pm 0.00098 \, \omega_0$ for the selectivity. In all these cases, the uncertainty is given for a confidence level of 95%. We see that the precision is several orders of magnitude better with the selectivity than with QFI, which is in agreement with the results of Sec. 5.2. We stress that the difference observed in this example is due to the choice of POVM. We have verified that if we increase the number of operators in the POVM such that it becomes informationally complete, the two controls give (up to small random fluctuations) the same results with the same precision.

The results obtained by the estimation process with the optimized QFI are, in the situation discussed here above, not the best one, but contrary to what we can expect from Fig. 3, the error is not infinite. The CFI computed with the corresponding control is constantly zero, but in fact, for $\Delta \neq \Delta_0$, the Bloch vector does not stay on the equator of the Bloch sphere, and at time $t_f$, we have a non-zero projection onto the $z$- axis. Therefore, we can estimate, in a limited way, the value of the parameter even if the CFI is zero. This effect is one of the non-local effects that play a role in the parameter estimation. This is welcome from the experimental point of view because the precision is better than expected, but it can also have a detrimental effect. Consider that the experimentally accessible POVM is compatible with the QFI (i.e. the CFI and the QFI are equal). For a closed quantum system, the maximum of QFI depends on time, and a better value can be reached for a longer control time. This argument is not valid for very large duration because the volume of the space of density matrices is finite. For a given final time, taken long enough, a given point of the space of density matrices can correspond to several values of $X$, and thus this may lead to a loss of precision in the measurement.

This can be easily observed with a spin-$\frac{1}{2}$ system. In Fig. 7, we show the ensemble of positions in the Bloch sphere, for a continuous ensemble of spin frequency $\Delta \in [-\omega_0, \omega_0]$, at time $t_f = 3\pi/\omega_0$, when the spins are driven by the optimal control maximizing the QFI for the offset $\Delta = 0$ (in this case, the control is a standard $\pi/2$ square pulse). For simplicity, we set $\alpha = 1$ and $\gamma = 0$ in the simulation. We observe that this ensemble of positions is a line on the sphere, that is self-intersecting at several positions. Hence, we would not be able to make the difference between the corresponding offset values if a measurement of the system state

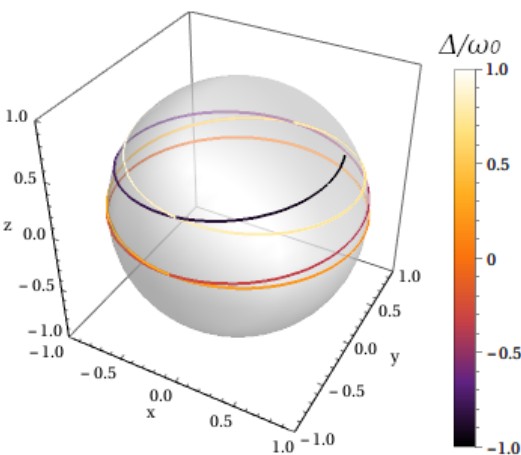

Figure 7: The line represents the positions in the Bloch sphere, at a fixed time $t_f$, of an ensemble of spin$-\frac{1}{2}$ with different offsets, when they all start their dynamics at the north pole of the sphere. The system Hamiltonian (in the rotating frame) is $H = -\frac{\Delta}{2}\sigma_z - \frac{\omega(t)}{2}\sigma_x$, with $\Delta \in [-\omega_0, \omega_0]$ the offset with respect to the rotating frame frequency, $\omega(t)$ the control field, and $\sigma_{x,z}$ two Pauli matrices. The control law is $\omega(t) = \omega_0$ if $t < \frac{\pi}{2\omega_0}$ and $\omega(t) = 0$ if $t \geq \frac{\pi}{2\omega_0}$. The final time is $t_f = \frac{3\pi}{\omega_0}$. We observe that the line is self-intersecting at 3 positions, given by $x = 0$ and $y \approx \pm 1$. Hence, a difference between the offset values cannot be made if a measurement of the system state returns one of these positions.

returns one of these positions. In practice, this situation is similar to the one already observed in Fig. 6 A. In the previous case, the effect was induced by a bad choice of POVM and it was not time-dependent, but in the present case, this happens only at specific times.

As a consequence, the fact that $X \mapsto \rho_X(t_f)$ is not injective in general must be taken into account a posteriori in the analysis of the control strategy used to maximize the QFI, i.e. we have to choose $t_f$ such that these problems cannot occur. On another side, this aspect can be avoided in the definition of the selectivity problem. Indeed, choosing $\delta X$ equal to the uncertainty of measurement of $X_0$ gives us a separation of quantum states well adapted to the experimental setup, and the non-injectivity problem of the mapping can thus be bypassed.

# 6 Conclusion

In this paper, we have compared two methods that can be used to design a control to improve parameter estimation of a quantum system. The first method is based on the maximization of Quantum Fisher Information at a fixed time, while the second approach consists of a time-optimal selective control problem. In the two cases, the underlying idea is to maximize the distance between two quantum states associated with two different values of the system parameters.

The two methods have several specificities that make them very different in practice. A summary of their differences and similarities is given in Tab. 1. Both techniques have their own advantages and difficulties. For QFI, it is easier to compute analytically the control or to have a global understanding of the optimal solution, at least in the case of a closed quantum system. However, the natural POVM of the QFI may not be adapted to the experiment, and we cannot easily take into account the non-local effects of the estimation problem. On the opposite, the selectivity method offers better flexibility in the computation of the control. Its

Table 1: Summary of the main elements of comparison between a control process maximizing the QFI and a time-optimal selective control. A cell in gray corresponds to an advantage of the corresponding method.

| | QFI maximization | Time-optimal selectivity |
|---|---|---|
| Local/Non-local | Local | Non-local |
| Intrinsic solution to the system | Yes | No |
| Suitable for experimental POVM | No | Yes |
| Intelligibility of the optimal solution | Easy to interpret | May be difficult to interpret |
| Taking into account the uncertainty on the parameter | With post-processing | Yes |
| Proof of optimality of the experimental uncertainty | Yes, but if the experimental POVM is adapted, and if $X_0$ is closed to the true value of $X$ | No |
| Equivalent to QFI maximization | - | yes, if the target states are well chosen |

main advantage is the fact that the target states can be adapted to the experimental POVM. We have restricted this study to two different methods, but we can imagine many slightly different approaches which mix several aspects of QFI and selectivity problems. For example, we can consider a version of the selectivity problem in which the target states are not fixed, the goal being to maximize the distance between the two states. The QFI and the selectivity methods can be seen as two complementary approaches. The QFI maximization can be used first, to highlight a specific control mechanism, which is used in a second step to compute a time-optimal selective control.

The Classical Fisher Information has been considered in this paper to characterize the control protocols. This quantity may also be used as a figure of merit to maximize. Using CFI instead of QFI may help avoid the POVM problem. We have not investigated this approach because the calculation of CFI is much more difficult than that of QFI, and even in simple cases, it is very difficult to obtain analytic or numerical control laws. The optimization algorithm stays generally stuck in a local minimum, which is not interesting for the estimation process. Therefore, selective control seems to be an interesting alternative to CFI maximization.

# Acknowledgments

We thank the group of Pr. D. Guéry-Odelin for many helpful discussions.

**Author contributions**   Q.A. and E.D. carried out the numerical simulations. Q.A. proposed and conducted the research project. All authors participated in the analysis of the results and in the paper redaction. All authors have read and agreed to the published version of the manuscript.

**Funding information**   This research has been supported by the ANR project "QuCoBEC" ANR-22-CE47-0008-02.

# A  Proofs of some mathematical theorems

## A.1  Limit properties between Bures and Fubini-Study metrics

In this appendix, we clarify the relation between Bures and Fubini-Study metrics which are different in the case where the support of the first-order density matrix is not included in the support of the unperturbed one. These points are already considered in [70]. They are described here for the sake of completeness, and with our own notations.

Our approach can be summarized as follows. First, a relation similar to the one of Thm. 1 is derived using the Bures metric as a starting point. The result is similar, but the sums do not run exactly in the same range. To be more specific, with the Bures distance, we are limited to the contributions included in the support of $\rho^{(0)}$, whereas the general formula contains contributions outside this support. Next, we consider the situation in which $\rho^{(0)}$ is full rank but approaches a pure state by means of a small parameter tending to zero.

We first focus on the calculation of $\lim_{\delta X \to 0} \frac{4}{(\delta X)^2} D(\rho_{X_0}, \rho_{X_0+\delta X})^2$. The non-trivial part concerns the computation of $\sqrt{\sqrt{\rho_1}\rho_2\sqrt{\rho_1}}$. Since the quantum fidelity is symmetric (and so is the Bures metric), we can choose $\rho^{(0)} \mapsto \rho_1$ and $\rho_{X_0+\delta X} \mapsto \rho_2$. Moreover, the definition of $\mathcal{F}$ indicates that a Taylor expansion of order 2 in $\delta X$ is needed. To this aim, we introduce the matrices $A$ and $B$ such that:

$$\sqrt{\sqrt{\rho^{(0)}}\rho_{X_0+\delta X}\sqrt{\rho^{(0)}}} = \rho^{(0)} + \delta X\, A + \delta X^2\, B + O(\delta X^3)\,. \tag{A.1}$$

Taking the square of this expression gives us:

$$\sqrt{\rho^{(0)}}\rho_{X_0+\delta X}\sqrt{\rho^{(0)}} = (\rho^{(0)})^2 + \delta X\left(\rho^{(0)}A + A\rho^{(0)}\right) + \delta X^2\left(\rho^{(0)}B + B\rho^{(0)} + A^2\right) + O(\delta X^3)$$

$$= (\rho^{(0)})^2 + \delta X\,\sqrt{\rho^{(0)}}\rho^{(1)}\sqrt{\rho^{(0)}} + \delta X^2\,\sqrt{\rho^{(0)}}\rho^{(2)}\sqrt{\rho^{(0)}} + O(\delta X^3)\,.$$

In the second line, we have used the Taylor expansion (3). Using the fact that $\sqrt{\rho^{(0)}} = \sum_k \sqrt{p_k^{(0)}}|\psi_k^{(0)}\rangle\langle\psi_k^{(0)}|$, and identifying the terms of the same power in $\delta X$, we obtain:

$$\langle\psi_k^{(0)}|A|\psi_l^{(0)}\rangle = \begin{cases} \dfrac{\sqrt{p_k^{(0)}p_l^{(0)}}}{p_k^{(0)} + p_l^{(0)}}\langle\psi_k^{(0)}|\rho^{(1)}|\psi_l^{(0)}\rangle\,, & \text{if } p_k^{(0)} \neq 0 \text{ and } p_l^{(0)} \neq 0\,, \\ 0\,, & \text{otherwise,} \end{cases} \tag{A.2}$$

$$\langle\psi_k^{(0)}|B|\psi_l^{(0)}\rangle = \begin{cases} \dfrac{\sqrt{p_k^{(0)}p_l^{(0)}}}{p_k^{(0)} + p_l^{(0)}}\langle\psi_k^{(0)}|\rho^{(2)}|\psi_l^{(0)}\rangle - \dfrac{1}{p_k^{(0)} + p_l^{(0)}}\langle\psi_k^{(0)}|A^2|\psi_l^{(0)}\rangle\,, & \text{if } p_k^{(0)}, p_l^{(0)} \neq 0\,, \\ 0\,, & \text{otherwise.} \end{cases} \tag{A.3}$$

Note that the matrix elements are 0 if $p_i^{(0)} = 0$ in the formula. This prevents any singularity or undetermined expression of the form 0/0. The trace is then given by

$$\mathrm{Tr}\left[\sqrt{\sqrt{\rho^{(0)}}\rho_{X_0+\delta X}\sqrt{\rho^{(0)}}}\right] = \mathrm{Tr}[\rho^{(0)}] + \delta X\,\mathrm{Tr}[A] + \delta X^2\,\mathrm{Tr}[B] + O(\delta X^3) \tag{A.4}$$

$$= 1 - \frac{\delta X^2}{2}\sum_{k|p_k^{(0)}>0}\frac{1}{p_k^{(0)}}\langle\psi_k^{(0)}|A^2|\psi_k^{(0)}\rangle + O(\delta X^3)\,. \tag{A.5}$$

The final result is drastically simplified because $\mathrm{Tr}[\rho^{(1)}] = \mathrm{Tr}[\rho^{(2)}] = 0$, since $\mathrm{Tr}[\rho_{X_0+\delta X}] = \mathrm{Tr}[\rho^{(0)}] = 1$. Note that this result holds only if the support of $\rho^{(n)}$, $n > 0$ is included in the support of $\rho^{(0)}$. Using Eq. (A.2) and the definition of the Bures metric (7), we arrive at:

$$D(\rho_{X_0}, \rho_{X_0+\delta X})^2 = \delta X^2 \sum_{k|p_k^{(0)}>0} \sum_{m|p_m^{(0)}>0} \frac{p_m^{(0)}}{(p_k^{(0)}+p_m^{(0)})^2} \left|\langle\psi_k^{(0)}|\rho^{(1)}|\psi_m^{(0)}\rangle\right|^2 + O(\delta X^3). \quad (A.6)$$

Plugging Eq. (A.6) into the definition of $\mathcal{F}$, given in Eq. (6), we arrive directly at Eq. (10), but with a difference on the sum over $k$.

Next, we proceed to the computation of $\mathcal{F}$ in the limit when the density matrix approaches a pure state. In this limit case, the support of $\rho^{(1)}$ is *not* included in the support of $\rho^{(0)}$.

We assume that the coefficients $p_k^{(0)}$ are of the form $p_k^{(0)} = \delta_{k0}(1 - a_0\epsilon) + (1 - \delta_{k0})\epsilon a_k$, with $\epsilon$ a small parameter such that $\lim_{\epsilon\to0}\rho^{(0)} = |\psi_0^{(0)}\rangle\langle\psi_0^{(0)}|$. The coefficients $a_0$ and $a_k$, which do not depend on $X$, are chosen such that $\mathrm{Tr}[\rho^{(0)}] = 1$. Next, we can compute the first order correction $\rho^{(1)}$. Using the fact that $\partial_X p_k^{(0)} = 0$ we have:

$$\rho^{(1)} = \sum_{k=0}^{dim\mathcal{H}-1} p_k^{(0)}\left(|\psi_k^{(0)}\rangle\langle\psi_k^{(1)}| + |\psi_k^{(1)}\rangle\langle\psi_k^{(0)}|\right). \quad (A.7)$$

Inserting Eq. (A.7) into Eq. (10), we arrive at:

$$
\begin{aligned}
\mathcal{F} &= 4\sum_{k,l,n} \frac{p_m^{(0)}p_n^{(0)}}{(p_m^{(0)}+p_k^{(0)})^2} \left|\langle\psi_k^{(0)}|\left(|\psi_n^{(0)}\rangle\langle\psi_n^{(1)}| + |\psi_n^{(1)}\rangle\langle\psi_n^{(0)}|\right)|\psi_m^{(0)}\rangle\right|^2 \\
&= 4\sum_{k,l} \frac{p_m^{(0)}}{(p_m^{(0)}+p_k^{(0)})^2} \left|p_k^{(0)}\langle\psi_k^{(1)}|\psi_m^{(0)}\rangle + p_m^{(0)}\langle\psi_k^{(0)}|\psi_m^{(1)}\rangle\right|^2 \\
&= 4\sum_{k,l} \frac{p_m^{(0)}(p_m^{(0)}-p_k^{(0)})^2}{(p_m^{(0)}+p_k^{(0)})^2} \left|\langle\psi_k^{(0)}|\psi_m^{(1)}\rangle\right|^2 .
\end{aligned} \quad (A.8)
$$

In the last equation, we use the relation $\langle\psi_k^{(0)}|\psi_m^{(0)}\rangle = \delta_{km}$, and thus $\langle\psi_k^{(1)}|\psi_m^{(0)}\rangle = -\langle\psi_k^{(0)}|\psi_m^{(1)}\rangle$. We can now work on the term $C_{km} = \frac{p_m^{(0)}(p_m^{(0)}-p_k^{(0)})^2}{(p_m^{(0)}+p_k^{(0)})^2}$ as a function of $\epsilon$. An explicit computation of the matrix elements gives us:

$$C = \begin{pmatrix} 0 & \epsilon a_1 & \epsilon a_2 & \\ 1-\epsilon(a0+4a_1) & 0 & \epsilon\frac{(a_1-a_2)^2 a_2}{(a_1+a_2)^2} & \cdots \\ 1-\epsilon(a0+4a_2) & \epsilon\frac{(a_1-a_2)^2 a_1}{(a_1+a_2)^2} & 0 & \\ & \vdots & & \ddots \end{pmatrix} + O(\epsilon^2). \quad (A.9)$$

We observe that in the limit $\epsilon \to 0$, only the terms $C_{k0}$ are non-zero and we have:

$$\lim_{\epsilon\to0}\mathcal{F} = 4\sum_{k>0} \left|\langle\psi_k^{(0)}|\psi_0^{(1)}\rangle\right|^2 . \quad (A.10)$$

This corresponds to the Fubini-Study metric, as given in Def. 1. To conclude, the Bures metric tends toward the Fubini-Study metric when the density matrix goes to a pure state, but the Bures metric is not equal to the Fubini-Study metric when it is evaluated with an exact pure state. This reveals a discontinuity in the Bures metric as discussed in [70].

## A.2  Proof of Thm. 2

*Proof.* The proof is established from the calculation of $|\psi^{(0)}(t)\rangle$ and $|\psi^{(1)}(t)\rangle$ using a time-dependent perturbation theory [84]. Explicitly, we have

$$|\psi(t)\rangle = U(t)|\psi_i\rangle = \left(U^{(0)}(t) + \delta X\, U^{(1)}(t)\right)|\psi_i\rangle + O(\delta X^2), \tag{A.11}$$

where $U(t)$ is the evolution operator from the initial time to time $t$, and $|\psi_i\rangle$ is the initial state, assumed to be independent of $X$. The first two orders of the Taylor expansion of the evolution operator with respect to $\delta X$ are given by:

$$U^{(0)}(t) = U(t)|_{X=X0}\,, \tag{A.12}$$

$$U^{(1)}(t) = -i\left[U^{(0)}(t)\int_0^t dt'\, U^{(0)\dagger}(t')\frac{\partial H(t')}{\partial X}U^{(0)}(t')\right]_{X=X0}. \tag{A.13}$$

Plugging the first-order term into Eq. (8) leads to:

$$\frac{\mathcal{F}}{4} = \langle\psi^{(0)}(0)|A^\dagger(t)A(t)|\psi^{(0)}(0)\rangle - \left|\langle\psi^{(0)}(0)|A(t)|\psi^{(0)}(0)\rangle\right|^2\,, \tag{A.14}$$

where

$$A(t) = \left[\int_0^t dt'\, U^{(0)\dagger}(t')\frac{\partial H(t')}{\partial X}U^{(0)}(t')\right]_{X=X0}. \tag{A.15}$$

The first term of Eq. (A.14) can be simplified by introducing a resolution of identity:

$$
\begin{aligned}
\langle\psi^{(0)}|A^\dagger(t)A(t)|\psi^{(0)}\rangle &= \int_0^t dt_1 \int_0^t dt_2 \langle\psi^{(0)}|U^{(0)\dagger}(t_1)\frac{\partial H(t_1)}{\partial X}U^{(0)}(t_1) \\
&\quad \times U^{(0)\dagger}(t_2)\frac{\partial H(t_2)}{\partial X}U^{(0)}(t_2)|\psi^{(0)}\rangle \\
&= \sum_{k=0}^{dim\mathcal{H}-1} \int_0^t dt_1 \int_0^t dt_2 \langle\psi^{(0)}|U^{(0)\dagger}(t_1)\frac{\partial H(t_1)}{\partial X}U^{(0)}(t_1)|k\rangle \\
&\quad \times \langle k|U^{(0)\dagger}(t_2)\frac{\partial H(t_2)}{\partial X}U^{(0)}(t_2)|\psi^{(0)}\rangle \\
&= \sum_{k=0}^{dim\mathcal{H}-1} \left|\int_0^t dt_1 \langle\psi^{(0)}|U^{(0)\dagger}(t_1)\frac{\partial H(t_1)}{\partial X}U^{(0)}(t_1)|k\rangle\right|^2.
\end{aligned}
$$

Note that we have used $|\psi^{(0)}\rangle = |\psi^{(0)}(0)\rangle$, to simplify the equations. Next, we can set $|\psi^{(0)}(0)\rangle = |\psi_0^{(0)}\rangle$ and $|k\rangle = |\psi_k^{(0)}\rangle$ without loss of generality, since we assume that the unperturbed eigenvectors are a basis of the Hilbert space. This allows us to remove the rightmost term in Eq. (A.14), and we finally arrive at:

$$\mathcal{F} = 4 \sum_{k>0}^{dim\mathcal{H}-1} \left|\int_0^t dt_1 \langle\psi_0^{(0)}|U^{(0)\dagger}(t_1)\frac{\partial H(t_1)}{\partial X}U^{(0)}(t_1)|\psi_k^{(0)}\rangle\right|^2 \tag{A.16}$$

$$= 4 \sum_{k>0}^{dim\mathcal{H}-1} \left|\int_0^t dt_1 \langle\psi_0^{(0)}(t_1)|\frac{\partial H(t_1)}{\partial X}|\psi_k^{(0)}(t_1)\rangle\right|^2. \tag{A.17}$$

$\square$

### A.3 Proof of Thm. 3

*Proof.* Our starting point is Eq. (8), which can be rewritten into Eq. (A.14). We notice that the QFI is proportional to the variance of $A$, since $A$ is hermitian. Then, we have

$$\mathcal{F} = 4\left\langle \left(A - \langle A\rangle_{\psi_0^{(0)}}\right)^2 \right\rangle_{\psi_0^{(0)}} = 4\left\langle \left(B - \langle B\rangle_{\psi_0^{(0)}}\right)^2 \right\rangle_{\psi_0^{(0)}}, \tag{A.18}$$

where in the second equality we have introduced $B = A - \text{eig min}(A)$, $\text{eig min}(A)$ denoting the smallest eigenvalue of $A$. Note that the spectrum of $B$ is such that: $0 \le \text{eig}(B) \le \text{eig max}(A) - \text{eig min}(A)$. We observe that

$$\left\langle (B - \langle B\rangle)^2 \right\rangle = \langle B^2\rangle - \langle B\rangle^2 \le c\langle B\rangle - \langle B\rangle^2, \tag{A.19}$$

with $c = \text{eig max}(B)$. This is a polynomial in $\langle B\rangle$ that can be maximized. A straightforward calculation gives the location of the maximum: $\langle B\rangle = c/2$. The maximum value is $c^2/2 - c^2/4 = c^2/4$. Therefore,

$$\mathcal{F} \le c^2 = (\text{eig max}(A) - \text{eig min}(A))^2. \tag{A.20}$$

Note that this computation is essentially a rewrite of a standard theorem in commutative probability theory, which can be found in [85]. The difference of eigenvalues of $A$ is straightforwardly obtained since it is the integral of an operator of the form $U^\dagger \partial_X H U$. The evolution operator gives us only a change of basis, and we have to consider the difference of eigenvalues of $\partial_X H$. Denoting them $\lambda_{\max}$ and $\lambda_{\min}$, we finally obtain

$$\mathcal{F} \le \left(\int_0^t dt_1 \, [\lambda_{max}(t_1) - \lambda_{\min}(t_1)]\right)^2. \tag{A.21}$$

$\square$

### A.4 Proof of Thm. 4

*Proof.* The starting point is the formula given in Eq. (10). We need to compute the coefficients $p_k^{(0)}$ and $\langle \psi_k^{(0)}|\rho^{(1)}|\psi_m^{(0)}\rangle$. We can set $p_k^{(0)} = \delta_{k0} + \epsilon p_k^{(1)}$ and $\rho^{(1)} = \rho^{(1,0)} + \epsilon\rho^{(1,1)}$. Then, we have

$$|\langle \psi_k^{(0)}|\rho^{(1)}|\psi_m^{(0)}\rangle|^2 = |\langle \psi_k^{(0)}|\rho^{(1,0)}|\psi_m^{(0)}\rangle|^2 + 2\epsilon\Re\left(\langle \psi_k^{(0)}|\rho^{(1,1)}|\psi_m^{(0)}\rangle\overline{\langle \psi_k^{(0)}|\rho^{(1,0)}|\psi_m^{(0)}\rangle}\right) + O(\epsilon^2). \tag{A.22}$$

We can go further by using

$$\rho(t) = U^{(0)}(t)\left\{\rho'(t) + i\,\delta X\,[\rho'(t), A(t)]\right\}U^{(0)}(t)^\dagger + O(\delta X^2),$$

with $\rho'(t) = |\psi_0\rangle\langle\psi_0| + \epsilon \sum_k p_k^{(1)}(t)|\psi_k\rangle\langle\psi_k|$. Inserting these expressions into Eq. (A.22), and using the notation $A_{km} = \langle \psi_k^{(0)}|A(t)|\psi_m^{(0)}\rangle$ gives us:

$$|\langle \psi_k^{(0)}|\rho^{(1,0)}|\psi_m^{(0)}\rangle|^2 = \left(|A_{0m}|^2\delta_{k0} + |A_{k0}|^2\delta_{m0}\right)(1 - \delta_{k0}\delta_{m0}), \tag{A.23}$$

and

$$\begin{aligned} &\Re\left(\langle \psi_k^{(0)}|\rho^{(1,1)}|\psi_m^{(0)}\rangle\overline{\langle \psi_k^{(0)}|\rho^{(1,0)}|\psi_m^{(0)}\rangle}\right) \\ &= \sum_n p_n^{(1)}\left\{|A_{0m}|^2\delta_{k0}\delta_{kn} - |A_{0n}|^2\delta_{mn}\delta_{k0} - |A_{n0}|^2\delta_{kn}\delta_{m0} + |A_{k0}|^2\delta_{m0}\delta_{mn}\right\}. \end{aligned} \tag{A.24}$$

Combining these expressions altogether, we get

$$
\begin{aligned}
\mathcal{F} &= 4 \sum_{k>0,m>0} \frac{p_m^{(0)}}{(p_m^{(0)}+p_k^{(0)})^2} \left( |A_{0m}|^2 \delta_{k0} + |A_{k0}|^2 \delta_{m0} \right) \\
&\quad + 8\epsilon \sum_{k,m,n} \frac{p_m^{(0)} p_n^{(1)}}{(p_m^{(0)}+p_k^{(0)})^2} \left( |A_{0m}|^2 \delta_{k0}\delta_{kn} - |A_{0n}|^2 \delta_{mn}\delta_{k0} - |A_{n0}|^2 \delta_{kn}\delta_{m0} + |A_{k0}|^2 \delta_{m0} \right) \\
&= 4\sum_{n>0} \frac{|A_{0n}|^2}{p_0^{(0)}+p_n^{(0)}} + 8\epsilon \sum_m \frac{|A_{0m}|^2}{(p_m^{(0)}+p_0^{(0)})^2} \left( p_m^{(0)}p_0^{(1)} - p_m^{(0)}p_m^{(1)} - p_0^{(0)}p_m^{(1)} + p_0^{(0)}p_0^{(1)} \right) \\
&= 4\sum_{n>0} \frac{|A_{0n}|^2}{p_0^{(0)}+p_n^{(0)}} + 8\epsilon \sum_m \frac{|A_{0m}|^2}{p_m^{(0)}+p_0^{(0)}} \left( p_0^{(1)} - p_m^{(1)} \right) \\
&= 4\sum_{n>0} |A_{0n}|^2 (1 - \epsilon(p_0^{(1)} + 3p_n^{(1)})),
\end{aligned}
$$

where in the last line we have used $(p_0^{(0)}+p_n^{(0)})^{-1} = 1 - \epsilon(p_0^{(1)}+p_n^{(1)}) + O(\epsilon^2)$. $\qquad\square$

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
