# Peer review of "Optimal Control Strategies for Parameter Estimation of Quantum Systems"

_SciPost Physics Core, doi:SciPost Phys. 16, 013 (2024)_

## Round 1 · Referee Report · Jing Liu · 2023-8-17

Report
The authors studied the performance of selective control processes in quantum parameter estimation and compared it with the direct maximization of the QFI/CFI. The topic is interesting and provides a useful angle for practical quantum metrology since different control algorithms or different controls may present different performances in different scenarios and a fair comparison would help the community better understand how to wisely chose the control amplitude or control itself in practice. I think this manuscript deserves to be published. My comments are as follows.
1) The cost function in Eq. (21) chooses a uniform distribution of the three functions F1, F2, and F12, is this uniform distribution optimal in this case or there exists a better distribution? In some cases, the optimal distribution might be learned via some machine learning techniques.
2) A typing error: in the caption of Figure 3, "d) Same as panel b)" should be "c) Same as panel b)".
3) Why does the performance between the optimal selective solution and the direct maximization of QFI/CFI show such a significant difference when the QFI is replaced with the CFI? In Fig. 3(b) the performance of the orange line is even a little worse than the dashed-blue line at the final time yet in Fig. 3(c), the orange line is significantly larger than the dashed-blue line at the final time.
4) Does the performance of Fig. 3(c) rely on the choice of POVM? Would it vanish when a different POVM is chosen?
5) The control amplitudes in Figs. 3 and 4 are discrete, and more importantly, the control amplitude number is small [in Fig. 3(d) there exist only 3 amplitudes for each control]. Do there exist some assumptions here or these controls are "globally" optimal? If there exist some assumptions on the amplitude number, the next question does the current results hold when the amplitude number is large? For example, the amplitude number is 100 instead of 3 in Fig. 3(d).
6) In the second paragraph of the introduction "we can mention the maximization of Quantum Fisher Information (QFI)", personally I think a related reference [Phys. Rev. A 96, 012117 (2017)] is missed.
Author: Quentin Ansel on 2023-09-18 [id 3988]
(in reply to Report 1 by Jing Liu on 2023-08-17)
We are grateful for this reviewing. We address in the following the points raised in the same order as they appear in the report. We have attached a version of the manuscript with changes written with colors.
1) The cost function in Eq. (21) chooses a uniform distribution of the three functions F1, F2, and F12, is this uniform distribution optimal in this case or there exists a better distribution? In some cases, the optimal distribution might be learned via some machine learning techniques.
Reply: We thank the Referee for this interesting comment. We consider in this section the QFI as a figure of merit. To study the properties of the control mechanism, we then show that this latter can be written as the sum of three different contributions denoted F1, F2 and F12. This expression of the cost functional C is given in Eq. (21) which is equivalent to the maximization of the QFI. As mentioned by the Referee, this second expression could be used as a new figure of merit (if the gradients of the different terms can be calculated analytically) by playing with the weights of the different contributions in order to enhance e.g. either the initialization or the stabilization processes. A different choice of cost functional will lead to a different control protocol. With non equal weights we loose the interpretation that C is the QFI a the final time, and thus, it is not easy to know for estimation parameter what the optimal distribution between the three functions could be. This point goes beyond the scope of this paper but could be an interesting direction to follow.
2) A typing error: in the caption of Figure 3, "d) Same as panel b)" should be "c) Same as panel b)".
The manuscript has been changed accordingly.
3) Why does the performance between the optimal selective solution and the direct maximization of QFI/CFI show such a significant difference when the QFI is replaced with the CFI? In Fig. 3(b) the performance of the orange line is even a little worse than the dashed-blue line at the final time yet in Fig. 3(c), the orange line is significantly larger than the dashed-blue line at the final time.
Reply:
- Concerning Fig 3(b): The optimal selective control has a slightly lower final QFI value because the spin state must leave the equator of the Bloch sphere, the equator being the ensemble of states maximizing the QFI (see paragraph below Eq. (26)). The QFI then increases with a sub-optimal rate at the end of the process. -Concerning Fig 3(c): The CFI is the quantity with the most important difference. Here, the optimal control of the QFI leads to a constantly zero CFI, which is due to the orthogonality of the optimal measure basis for $\Delta$ and the basis $|\uparrow\rangle,|\downarrow \rangle$. However, with the optimal selective control, we reach the maximum value of the CFI at the very end of the control process, because the target states correspond to the measurement basis." These points are discussed in p. 13. We have modified slightly the text to emphasize certain key points.
4) Does the performance of Fig. 3(c) rely on the choice of POVM? Would it vanish when a different POVM is chosen?
Reply:
We confirm the Referee’s intuition on the fact that the choice of POVM has an impact on the time evolution of the CFI. We could have chosen a POVM such that the dashed blue curve in Fig. 3c is different from zero. For example, in the example of section 5.3, a single curve is represented in each plot because the results are similar in all cases. This observation is due to the fact that the POVM is well adapted to the different situations under study.
5) The control amplitudes in Figs. 3 and 4 are discrete, and more importantly, the control amplitude number is small [in Fig. 3(d) there exist only 3 amplitudes for each control]. Do there exist some assumptions here or these controls are "globally" optimal? If there exist some assumptions on the amplitude number, the next question does the current results hold when the amplitude number is large? For example, the amplitude number is 100 instead of 3 in Fig. 3(d).
Reply: In the case of Figure 3, the controls are globally optimal. This is discussed in the lower part of p. 12 and in the upper part of p. 13. We do not proceed to an exhaustive justification of the global optimality because most of the key points have been already discussed in the literature, in a different context. We refer to [45,46,32] for the technical details. In Ref. [33], some of the authors provide an exhaustive analysis of the optimality of the selective control field of Fig 3(d) in the case without relaxation. Since the control fields are globally optimal, increasing the number of pulses cannot further increase the efficiency of the control processes. In Fig. 4, we agree with the Referee that a specific constraint has been added to the control search. We assume that the control is a piecewise constant function with five-time steps in which the phase, the amplitude and the duration of the time step are optimized. As mentioned in the paper, this choice leads to a good compromise between computational time and control efficiency. Numerical optimization methods with smooth controls could also be used to solve this control problem, but with final fidelities of the same order of magnitude.
6) In the second paragraph of the introduction "we can mention the maximization of Quantum Fisher Information (QFI)", personally I think a related reference [Phys. Rev. A 96, 012117 (2017)] is missed.
Reply: We thank the Referee for pointing out this reference which has been added to the bibliography of the paper.
Author: Quentin Ansel on 2023-09-18 [id 3989]
(in reply to Report 2 on 2023-09-05)We are grateful for this reviewing. We address in the following the points raised in the same order as they appear in the report. We have attached a version of the manuscript with changes written with colors.
1) The connection between QFI and the Bures distance is only valid locally for neighboring states where D(ρx, ρx+δx) << 1, using Eq.(16) in the regime D2 = 2 for the connection to selective control is a bit problematic.
Reply. We disagree with this comment of the Referee. In Eq. (16), we introduce \mathcal{F}_{fd}, which is a finite difference approximation of the QFI. The QFI is reached only when the difference between the two density matrices goes to zero. This approximation is precisely used to avoid mathematical problems. With this finite difference version of the QFI, it is straightforward to see that the QFI is necessarily infinite if the goal is to generate orthogonal states for two infinitesimally close system parameters. With the assumption that the QFI increases monotonically with time, this situation can only be achieved in infinite time.
2) In the case of fixed POVM, the comparison of QFI based optimization and the selective control seems unfair. Here the relevant quantity is CFI, which may be connected to the selectivity of the probability distributions
Reply. We completely agree with this comment of the Referee mentioning that the relevant quantity is the CFI. We have commented on our choice in detail in several paragraphs (see e.g. p 6 above Def. 3, in Sec. 5.5, and in the conclusion). We have focused our attention on the QFI, for the following reasons: - Many different studies (see the literature presented in the introduction) investigated the optimal control of the QFI, and it was interesting to revisit this point and to show the limits of this approach. - The optimization of the CFI with a control field is in general a very arduous numerical problem, due to numerical instabilities and control traps, while the maximization of the QFI is easier. It is therefore very interesting to consider the second problem rather than the first and to check a posteriori whether the solution is indeed relevant experimentally. The complete analysis of the connection between CFI and selectivity goes beyond the scope of this paper and will be done in a forthcoming paper.
3) The performance of the selective control for parameter estimation depends on the choices of the target states. It is not clear how these states should be chosen. Even for the specific qubit example for the estimation of γ presented in the manuscript, the choice of the target state as the completely mixed state is not well explained. It is not clear why the other pole is not used as the target state. Even the steered state cannot reach it, why it is not chosen so the steered state can be made as closer to it as possible?
Reply. We confirm the comment of the Referee in the sense that there is no recipe for the choice of target state, this is left to the intuition of the physicist based on knowledge of system dynamics. This is one of the main difficulties of this approach, as outlined in the conclusion (the solution is not intrinsic to the system). For the estimation of γ, we have clarified this point at the end of page 16. We consider the north pole as one of the target states because it is the attractor of the relaxation process, ensuring that a final state close to this target can be reached. For the second state, we do not take the opposite pole because we know that it cannot be reached (due to relaxation effects, the reachable set corresponds only to a subspace of the Bloch ball. An interesting reference on the subject is: PhysRevA.88.033407). Instead, we consider an intermediate position with the center of the Bloch ball. We could have used other target states for which the result would have been very similar. Here, the positions of the target states are not a key factor because orthogonal states cannot be generated, due to relaxation process.
4) As briefly mentioned in the conclusion, the QFI has a closer connection to the selective control when the target states are not fixed, but directly maximizing the distance of the final states. With fixed target states, it seems the two methods are equivalent only when D(ρ0, ρtarget,0) +D(ρ1, ρtarget,1) = D(ρ0, ρ1) where ρ0 and ρ1 are the final states. For the examples where the selectivity defers from the QFI, I would suggest the authors to consider checking this condition and if possible choose a different set of target states that satisfy this condition for a further comparison.
Reply. We agree with this comment of the Referee, but we must be careful when comparing the two variants of the selectivity problem. If the target states are fixed, we minimize a cost functional of the form D(ρ0, ρtarget,0) +D(ρ1, ρtarget,1), while if the targets are free, we maximize D(ρ0, ρ1). After optimization, the first cost functional is ideally equal to 0. In this second case, the results using a selective control are expected to be closer to an optimization with the QFI.
Attachment:
diff_v1_to_v2_omCzNda.pdf

---

## Round 1 · Referee Report · Anonymous · 2023-9-5

In this manuscript the authors compare two different methods: the QFI maximization and the time-optimal selectivity. The former aims to maximize the quantum Fisher information, while the latter seeks to optimize the selectivity of the control. The authors provide a concrete example of the estimation of the values of the Hamiltonian parameters of a spin-1/2 system coupled to a bosonic reservoir at zero temperature. They compare the two methods using the Bures distance, the QFI, and the CFI. The authors conclude that the QFI and the selectivity methods can be seen as complementary approaches to the same problem, and that the selectivity method offers better flexibility in the computation of the control. The manuscript provides some interesting observations on the similarities and differences between two different methods but there are a few points that the authors need to clarify:

1. The connection between QFI and the Bures distance is only valid locally for neighboring states where $D(\rho_x, \rho_{x+\delta x}) << 1$, using Eq.(16) in the regime $D^2 = 2$ for the connection to selective control is a bit problematic.

2. In the case of fixed POVM, the comparison of QFI based optimization and the selective control seems unfair. Here the relevant quantity is CFI, which may be connected to the selectivity of the probability distributions.

3. The performance of the selective control for parameter estimation depends on the choices of the target states. It is not clear how these states should be chosen. Even for the specific qubit example for the estimation of $\gamma$ presented in the manuscript, the choice of the target state as the completely mixed state is not well explained. It is not clear why the other pole is not used as the target state. Even the steered state can not reach it, why it is not chosen so the steered state can be made as closer to it as possible?

4. As briefly mentioned in the conclusion, the QFI has a closer connection to the selective control when the target states are not fixed, but directly maximizing the distance of the final states. With fixed target states, it seems the two methods are equivalent only when $D(\rho_0, \rho_{target,0}) + D(\rho_1, \rho_{target,1}) = D(\rho_0, \rho_1)$ where $\rho_0$ and $\rho_1$ are the final states. For the examples where the selectivity defers from the QFI, I would suggest the authors to consider checking this condition and if possible choose a different set of target states that satisfy this condition for a further comparison.

---

## Round 2 · Referee Report · Anonymous (Referee 3) · 2023-9-23

Report

I think the revision has well included all my concerns, I would like to recommend it to be accepted.

---

## Round 2 · Referee Report · Anonymous (Referee 4) · 2023-10-7

Report

I am a bit confused by the author's response to the first comment. On one hand, the authors disagree with the comment, and on the other hand, it is stated that "The QFI is reached only when the difference between the two density matrices goes to zero", which agrees with the comment. Do the authors mean although they use orthogonality as the objective it is never achieved and the distance of the states remains small? This does not agree with the examples of the manuscript where the distance is not necessarily infinitesimally small.

In practice, it is fine to use the finite difference as a heuristic cost function for the optimization, but it should be kept in mind that this can differ from the QFI. It is fine to write the finite difference as $F_{fd}(t_f) =\frac{8}{\delta X^2}$ but assume it equals to the QFI and write $F=\frac{8}{\delta X^2}$, as if the QFI depends on $\delta X$, is confusing. Actually, the authors have observed the differences as the Bures distance and the QFI do not coincide with each other in some examples in the manuscript.

For the third point on why the other pole is not chosen as the other objective state, the authors did not fully address the question: "Even the steered state cannot reach it, why it is not chosen so the steered state can be made as close to it as possible?". An objective state does not necessarily have to be reached, it just sets a target that steers the state as close to it as possible. It is not quite clear why the orthogonal states cease to be good target states when they can not be reached.

  • validity: good
  • significance: good
  • originality: good
  • clarity: high
  • formatting: excellent
  • grammar: excellent

Author:  Quentin Ansel  on 2023-10-27  [id 4071]

(in reply to Report 2 on 2023-10-07)
Category:
answer to question

We thank the Referee for these interesting comments.

1) We almost agree with the opinion of the Referee but there are some confusions, which may arise from our previous reply. We therefore start the explanation from the beginning to clarify the different points mentioned by the Referee.

Consider the finite difference version of the QFI: $\mathcal F_{fd} = 4 D/\delta X^2$. This quantity strictly reaches the QFI when $\delta X$ tends to zero. Now, suppose that for some $\delta X$, it is possible to freely set the value of D (i.e., we have complete controllability of the system). Then we can choose D^2=2 which corresponds to the maximum value of D, and we have $ F_{fd} = 8/\delta X^2$. However, the controls that allow us to generate orthogonal states have an increasing duration when \delta X decreases. Consequently, for $\delta X$ -> 0, the control time goes to infinity, and in practice, we cannot produce orthogonal states in a finite time for two infinitesimally close values of X.

We can reverse the reasoning. We can set a finite value of $F$, and search for a value of $\delta X$ such that $F_{fd}$ is very close to $F$. For $\delta X$ small enough, we are sure that such a situation can happen. Thus, in specific situations, we can expect that both QFI and selective optimizations lead to the same (or at least, similar) control process, and the result should be almost identical with both approaches. This is basically the message given in pages 8 and 9 of the manuscript. Note that in Eq.~(17), we have written $\mathcal F \simeq \frac{8}{\delta X^2} $ and not $\mathcal F= \frac{8}{\delta X^2}$, to specify that the quantities are not strictly equivalent.

However, there are some very specific cases in which $\mathcal F_{fd} =\mathcal F$ for a non-zero value of $\delta X$. This is the case when the optimization process that generates orthogonal states for a non-zero value of $\delta X$ coincides with the optimal control of the QFI (see examples Sec. 5.3 and 5.4). Such a situation can be described as follows: Set a value $\mathcal F$ that must be obtained at time $t$, choose $\delta X$ such that $\mathcal F= \frac{8}{\delta X^2}$, and find the control that generates $D^2 = 2$ in time $t$. Of course, this reasoning must be taken with caution because the problem may not have a solution. We present a counter-example in Fig. 4 where the Bures distance and the QFI do not agree very well. We are in this case outside the domain of validity in which $\mathcal F_{fd}$ is very close to $\mathcal F$.

2) We agree with the second comment of the Referee: " An objective state does not necessarily have to be reached; it just sets a target that steers the state as close to it as possible". We could have chosen orthogonal states for this optimization procedure. It is likely that, in this situation, the result would not have been very different. However, since we know that the maximum of the cost functional cannot be reached, the result may be suboptimal with respect to the goal of increasing the distance between the two systems. We therefore prefer in this case not to use orthogonal states. Numerical simulations show that the South Pole is a good choice of target state.

We modified the text to clarify these points. We have attached a version of the manuscript with changes in red characters.

Attachment:

Article_QFI_sélectivité_v2-1.pdf

---

## Round 2 · Author Response

Dear Editor,

Please find herewith a revised version of the manuscript entitled ``Optimal control strategies for parameter estimation of quantum systems" that we would like to resubmit for publication in SciPost Physics

The two referees report a positive judgment of our work. The two referees raise several interesting questions and point out different technical points to clarify. We have taken into account, in the new version of the manuscript, the different comments of the referees.

In the reply of each report, we have included a revised version of the manuscript with changes written with colors. We hope that these comments and clarifications will render this article suitable for publication in SciPost Physics.

Yours sincerely,
the authors

---

## Round 2 · List of Changes

- p 2. New reference.
- p 13. New sentence: "we recall that
the equato ris the set of states maximizing the increase of QFI"
- p 16-17. New paragraph : "This is due to... pole of the Bloch sphere".
- p 17. Modification of a sentence: "We observethat..."

---

## Round 3 · Referee Report · Anonymous (Referee 5) · 2023-11-3

Report

I appreciate the authors' clarification but further clarifications on the relationships between $F_fd$ and F may needed to avoid potential confusions:
1) It is right that making $\delta x$ 'sufficiently small' $F_{fd}$ can approach $F$. But 'sufficiently small' typically means the distance between the two states needs to be sufficiently small, i.e., D is sufficiently small. Since F corresponds to the second order expansion and typically $\delta x t$ goes into the expansion, so $\delta x t$ together needs to be suffiently small to make the second order expansion valid.
2)For the procedure that first choosing a \delta x such that $F_{target}=8/\delta x^2$ then find a control to make $D^2=2$, this procedure makes $F_{fd}=F_{target}$, but does not make $F_{fd}$ equal to the real $F$, since $F_{target}$ is just an arbitrary number set before the control and the final state, it does not equals to the QFI of the actual final state---unless $F_{fd}=F$ which goes back to the original point .
As I said, it is fine to use $F_{fd}$ as the figure of merit, but the authors should avoid possible confusions.

  • validity: high
  • significance: good
  • originality: good
  • clarity: good
  • formatting: excellent
  • grammar: excellent

Author:  Quentin Ansel  on 2023-11-23  [id 4143]

(in reply to Report 1 on 2023-11-03)
Category:
reply to objection

We thank the Referee for this interesting comment. We agree that it is important to clarify the different quantities used in the paper. Following the comments, the text has been modified to highlight the differences between $\mathcal F$ and $\mathcal F_{fd}$. In particular, the comment in the footnote on page 9 is now inserted in the main text as follows:
\begin{equation}
\mathcal F \simeq \frac{8}{\delta X^2} = 8 \alpha t_{\textrm{min}}^2,
\end{equation}
with $\alpha$ a constant specific to the system. When $\delta X \rightarrow 0$, we obtain $t_{\textrm{min}} \rightarrow \infty$ and $\mathcal F \rightarrow \infty$. This result shall be manipulated with caution because $\mathcal F_{fd}$ gives us only an approximation of $\mathcal F$. However, in some cases, the optimization of the two quantities can lead to the same result. This is the case when the optimization process which generates orthogonal states for a non-zero value of $\delta X$ coincides with the optimal control of the QFI.

With this new formulation, we emphasize that $\mathcal F_{fd}$ is not necessarily equal to $\mathcal F$, but the optimization of $\mathcal F_{fd}$ can help maximizing $\mathcal F$. This point can be justified qualitatively from a Taylor expansion of the function and an exact treatment of the reminder. Consider for instance a function $f$ such that $f(0) = df/dx(0) =0$ (this is the case for the QFI). Then,
\[
\exists c \in ]0,b[~|~\frac{d^2f}{dx^2}(c) = \frac{f(b)}{2 b^2}
\]
Note the strict equality between the second derivative of the function at x=c and the function itself at x=b. Here $d^2f/dx^2(0)$ and $f(b)$ play respectively the role of $\mathcal F$ and $\mathcal F_{fd}$. For a fixed value of $b$, chosen small enough, we deduce that maximizing $f(b)$ also amounts to maximizing $d^2f/dx^2(c)$. If the variations of the second derivative $d^2f/dx^2$ are not too strong in the small interval $[0,c]$, this also amount to maximizing $d^2f/dx^2(c)$. The same kind of argument can be used in the paper for the QFI. This analysis in terms of Taylor expansion is not discussed in depth in the manuscript because it requires careful treatment of the logarithmic derivative operator $L$ (to be mathematically well justified). A brief footnote comment has been added in page 9, to clarify this point.

---

## Round 3 · Author Response

Dear Editor,

Please find herewith a second revised version of the manuscript entitled ``Optimal control strategies for parameter estimation of quantum systems" that we would like to resubmit for publication in SciPost Physics

The first Referee has accepted the publication of this manuscript. Additional questions are raised by the second Referee. You can in our reply the responses to these new comments raised by the second Referee.

We hope that these comments and clarifications will render this article suitable for publication in SciPost Physics Core. We have also corrected some misprints that we have detected in the text.

Yours sincerely,
the authors

---

## Round 3 · List of Changes

List of changes :

-page 9 : « (orthogonal states may be generated only in infinite time for two infinitesimally close parameters) »

-page 9 : footnote, « $\Mc F_{fd}$ gives us only an approximation of $\Mc F$, but in some situations, the optimization of the two quantities can lead to the exact same result. This is the case when the optimization process that generates orthogonal states for a non-zero value of $\delta X$ coincides with the optimal control of the QFI. Several examples of such a situation are given in Sec.5. »

- page 17 : « These target states are not orthogonal, but they avoid the search for suboptimal solutions with respect to the goal of increasing the distance between the two systems. »

---

## Round 4 · Referee Report · Anonymous (Referee 2) · 2023-11-30

Report

I recommend the publication.

---

## Round 4 · Author Response

Dear Editor,
Please find herewith a third revised version of the manuscript entitled ``Optimal control strategies for parameter estimation of quantum systems" that we would like to resubmit for publication in SciPost Physics Core

The first Referee has accepted the publication of this manuscript. Additional questions are raised by the
second Referee. We changed the text to meet the Referee's request and made it clearer that $\mathcal F_{df}$ and
$\mathcal F$ do not always coincide.

We hope that these comments and clarifications will render this article suitable for publication in SciPost Physics Core. We
have also corrected some misprints that we have detected in the text.

Yours sincerely,

the authors

---

## Round 4 · List of Changes

• Modification of the footnote p.9 (see reply to the referee for details)
  • Insertion of a new sentence below Eq. (17): "This result shall be manipulated with caution because $\Mc F_{fd}$ gives us only an approximation of $\Mc F$. However, in some cases, the optimization of the two quantities can lead to the same result."

---

## Editorial Decision

published